# Downscaling epidemiological time series data for improving forecasting accuracy: An algorithmic approach

**Mahadee Al Mobin**[1,2], **Md. Kamrujjaman**[1] *

**1** Department of Mathematics, University of Dhaka, Dhaka, Bangladesh, **2** Bangladesh Institute of Governance and Management, Dhaka, Bangladesh

* kamrujjaman@du.ac.bd

## Abstract

Data scarcity and discontinuity are common occurrences in the healthcare and epidemiological dataset and often is needed to form an educative decision and forecast the upcoming scenario. Often to avoid these problems, these data are processed as monthly/yearly aggregate where the prevalent forecasting tools like Autoregressive Integrated Moving Average (ARIMA), Seasonal Autoregressive Integrated Moving Average (SARIMA), and TBATS often fail to provide satisfactory results. Artificial data synthesis methods have been proven to be a powerful tool for tackling these challenges. The paper aims to propose a novel algorithm named Stochastic Bayesian Downscaling (SBD) algorithm based on the Bayesian approach that can regenerate downscaled time series of varying time lengths from aggregated data, preserving most of the statistical characteristics and the aggregated sum of the original data. The paper presents two epidemiological time series case studies of Bangladesh (Dengue, Covid-19) to showcase the workflow of the algorithm. The case studies illustrate that the synthesized data agrees with the original data regarding its statistical properties, trend, seasonality, and residuals. In the case of forecasting performance, using the last 12 years data of Dengue infection data in Bangladesh, we were able to decrease error terms up to 72.76% using synthetic data over actual aggregated data.

## Introduction

Any process that involves deriving high-resolution data from low-resolution variables is referred to as downscaling. This method relies on dynamical or statistical approaches and is extensively utilized in the field of meteorology, climatology, and remote sensing [1, 2]. Significant exploration of the downscaling methods has been done in the field of geology and climatology to enhance the out of existing models like the General Circulation Model (GCM) [3–8], Regional Climate Model (RCM) [9], Integrated Grid Modeling System (IGMS) [10], System Advisor Model (SAM) [10] and to make it usable for the forecast of geographically significant region and time. Several methods has been used to downscale these data such as BCC/RCG-Weather Generators (BCC/RCG-WG) [11–13], and Statistics Downscaling Model (SDSM) [11, 14–19], Bayesian Model Averaging (BMA) [20]. Even machine

**Data Availability Statement:** All relevant data are within the paper and its Supporting Information files.

**Funding:** The author(s) received no specific funding for this work.

**Competing interests:** The authors have declared that no competing interests exist.

learning methods has been used like Genetic algorithm (GA) [9], K Nearest Neighbourhood Resampling (KNNR) [9], Support Vector Machine (SVM) [11, 21–23]. Except for the machine learning algorithms, which are methods that are finding their applications in new domains, the rest of the methods are tailored to suit the outputs of the models, as mentioned earlier.

This class of methods has recently been applied in the disaggregation of spatial epidemiological data [24, 25]. Nevertheless, significant work has yet to be done for the temporal downscaling of epidemiological data. The temporal downscaling techniques are often classical interpolation techniques that do not do justice to aggregated data. This phenomenon can be well illustrated with an example. Consider the case of monthly Dengue infection data of 2017 from Fig 1, which has been downscaled using linear interpolation by considering the aggregated value as the value of the end date of a month in Fig 2. In this case, if we consider the monthly aggregate of the downscaled data, it does not match the original aggregate. This downscaled data, which differs from the original data in such statistical measures, shall result in decisions and knowledge that cannot be far from the truth.

The paper aims to achieve the following:

- To propose a novel algorithm named Stochastic Bayesian Downscaling (SBD) algorithm based on the Bayesian approach that can regenerate downscaled temporal time series of varying time lengths from aggregated data preserving most of the statistical characteristics and the aggregated sum of the original data.

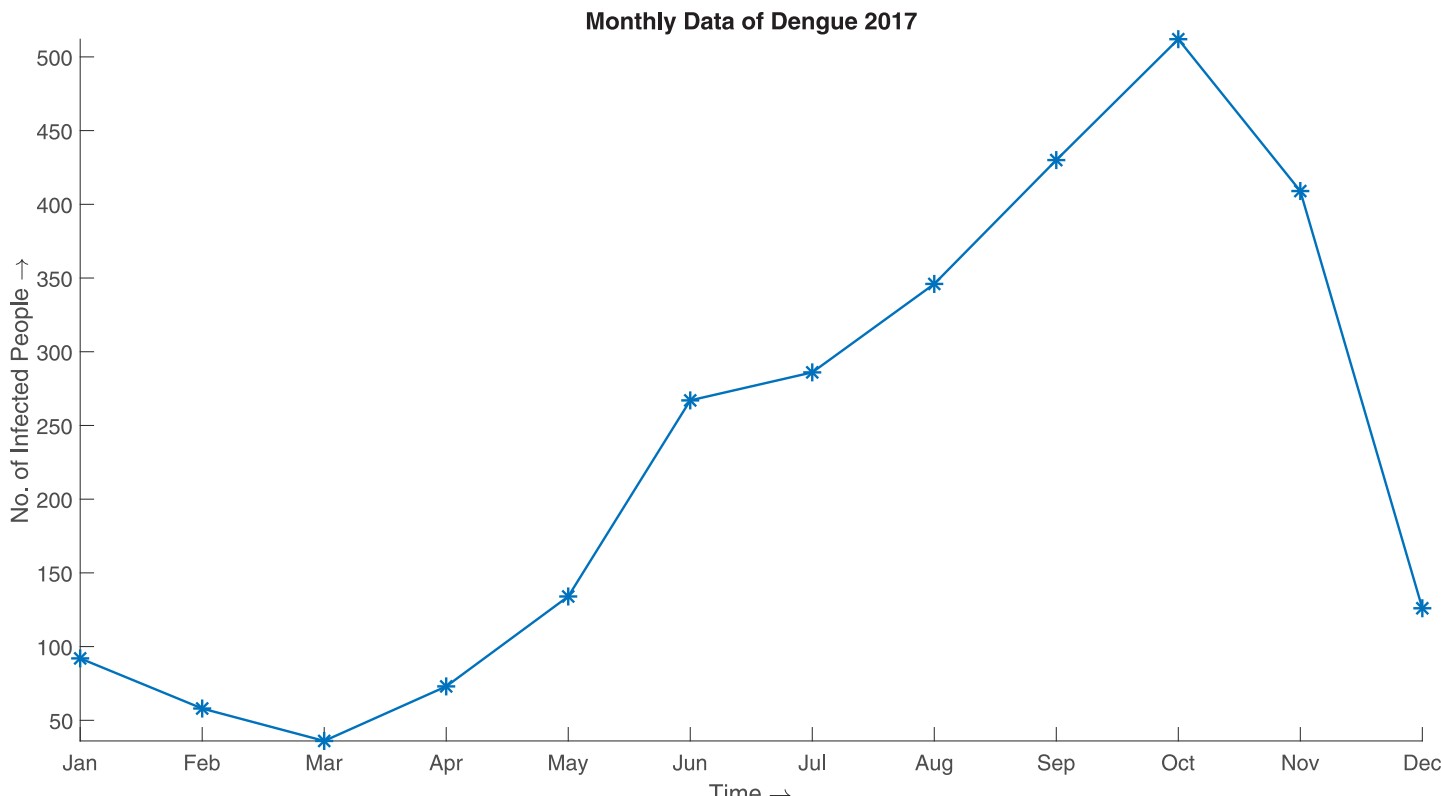

**Fig 1. Monthly data of Dengue 2017.** The monthly aggregate of the DENV infection in Bangladesh in the year 2017. The data has been aggregated to monthly scale to avoid the discontinuity observed in the daily scale.

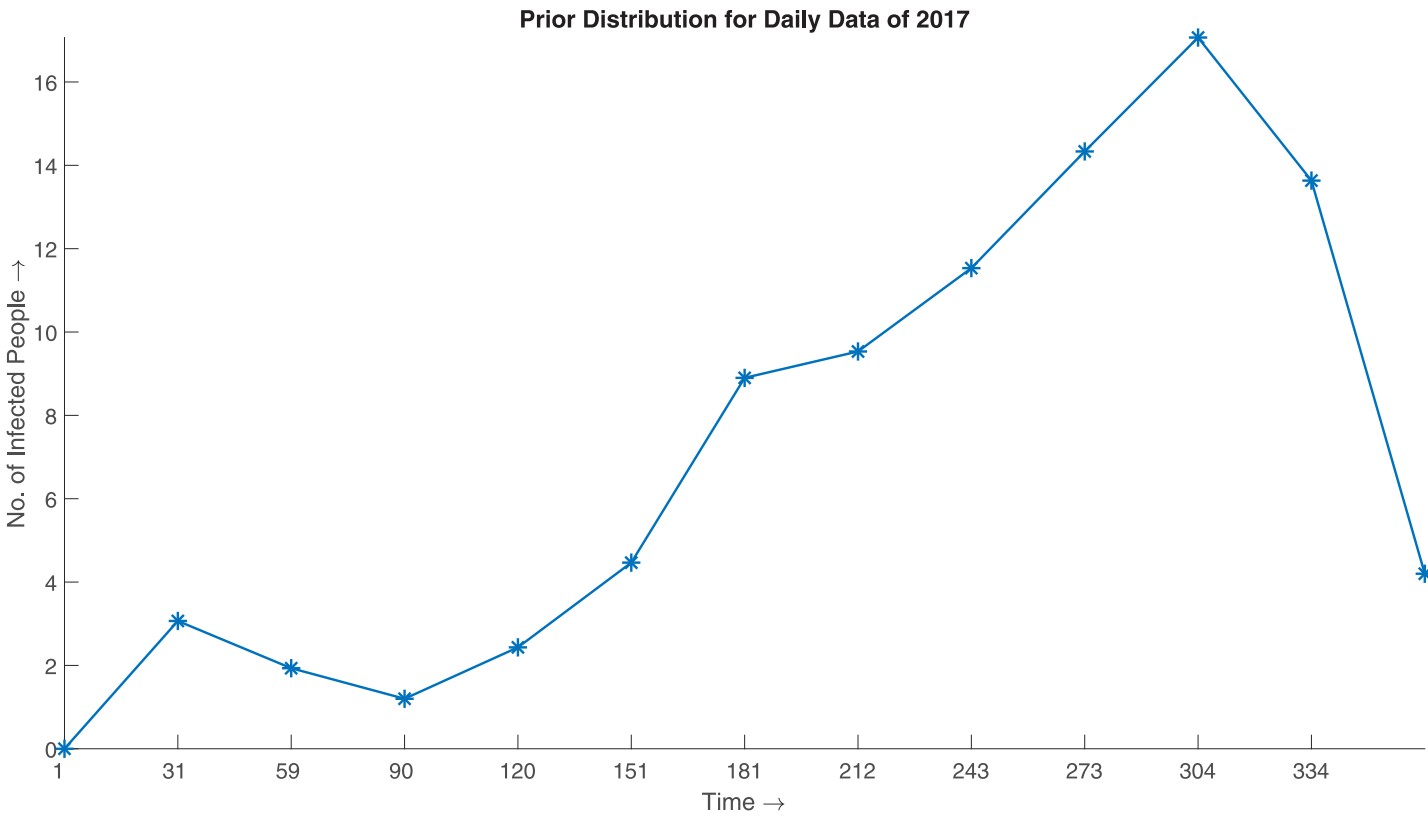

**Fig 2. Prior distribution for daily data of 2017.** The figure depicts the downscaled data using linear interpolation by considering the aggregated value as the value of the end date of a month using the data illustrated in Fig 1. In this case, if we consider the monthly aggregate of the downscaled data, it does not match the original aggregate. This downscaled data, which differs from the original data in such statistical measures, shall result in decisions and knowledge that cannot be far from the truth.

- To present two downscaling case studies of epidemiological time series data (namely Dengue and COVID-19 data of Bangladesh) to showcase the workflow and efficacy of the algorithm.

- To present a comparison between the forecasting performance of aggregated data and algorithm generated synthetic data to showcase the improvement achieved (for synthetic data over aggregated data) in terms of scale independent error.

The paper is organized as follows. Materials and method section describes the data used for the paper and its sources and the methodology at length with the proposed SBD algorithm. The section titled "Comparison of the Synthesized Data with the Real Data" compares the synthesized data with the actual data of two different epidemiological cases (Dengue and COVID-19) in Bangladesh and shows how the SBD algorithm could generate statistically accurate approximate of the actual with very little input in both cases discuss the benchmark metric used for evaluating the output. Section titled "Improvements in Forecasting Accuracy" shows the improvement of the forecasting accuracy using synthesized data over aggregated data using a statistical forecasting toolbox in the dengue scenario of Bangladesh using the last 12 years of monthly aggregated data, Forecasting model selection procedures, and residuals. Finally, we concludes our paper with an overview of the paper and how our paper has contributed to the existing literature and scopes for improvements and fields of application of the SBD algorithm in the conclusion section.

## Materials and methods

### Data

The dengue data from Bangladesh used in this paper are from January 2010 to July 2022 and are collected from DGHS [26], and IEDCR [27]. The COVID-19 data of Bangladesh are from 8 March 2020 to December 2020 and are collected from the WHO data repository [28].

### Methodology

The SBD algorithm can be segmented into three sequential parts, as exhibited in Fig 3. Initially, the algorithm considers a prior distribution to generate synthetic downscaled data. The SBD algorithm considers the aggregated data as the prior distribution of the downscaled data. For example: If we have the monthly epidemiological data of dengue for the year 2017, thus to attain the prior distribution for the downscaled data, we divide the data by 30. The fact is well illustrated in Figs 1 and 2. Fig 1 depicts the monthly distribution of the DENV (Dengue Virus) infection in Bangladesh for the year 2017, and Fig 2 represents the prior distribution obtained by the method described above.

Based on the prior distribution, initial statistical properties of the synthetic data are obtained except for the standard deviation ($\sigma$). As $\sigma$ is scaling independent, hence scaling method used to obtain the prior distribution from the monthly aggregate keeps the $\sigma$ identical to the monthly aggregate. To overcome this problem, we consider,

$$\sigma_0 = \frac{\sigma_{prior\ distribution}}{30} \qquad (1)$$

Flow Diagram of Stochastic Bayesian Downscaling Algorithm

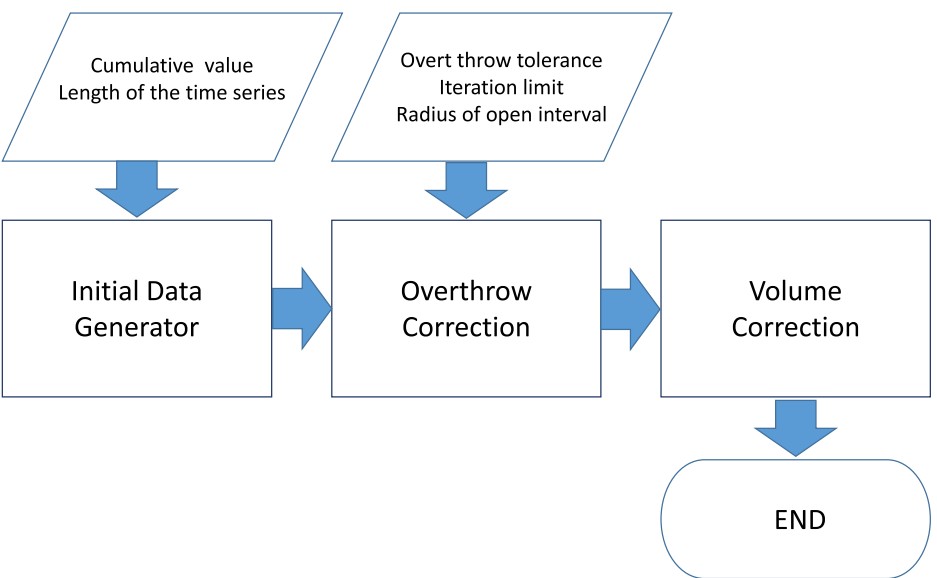

**Fig 3. Flow diagram of Stochastic Bayesian Downscaling algorithm.** The diagram depicts the flow diagram of the novel proposed algorithm. The algorithm works in three unique phases. The first phase (Initial Data Generator) generates a initial approximation based on the prior distribution, the second phase (Overthrow Correction) removes any abrupt fluctuation introduced during the de-aggregation in the first step, and finally the final step(Volume Correction) rectifies the any displacement of data point volume over the aggregation unit in the second step thus aggregation of this downscaled data agrees with the initial data.

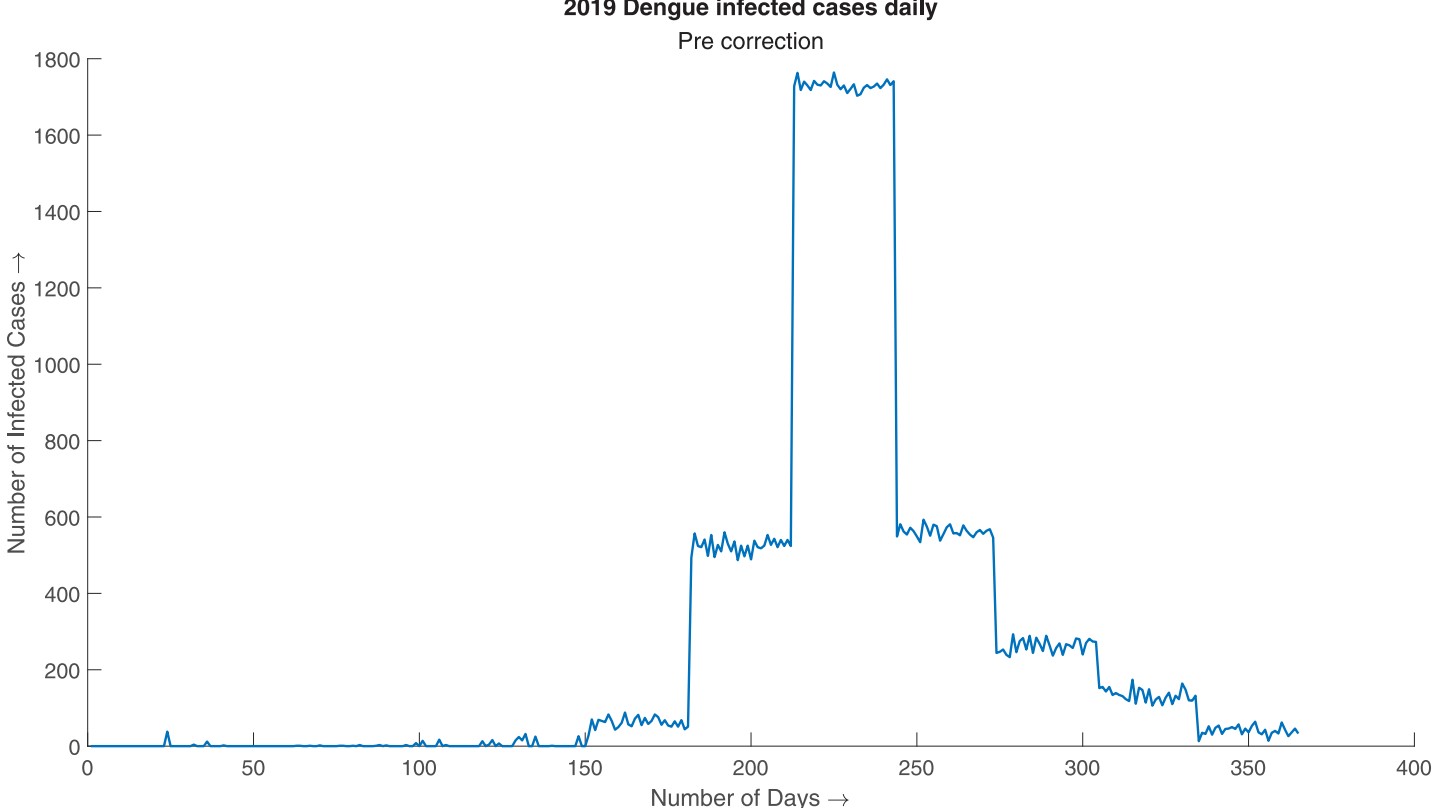

**Fig 4. 2019 Dengue infected cases daily, pre correction.** Initial approximation without overthrow correction exhibits a staircase like property due to higher gradient change of the prior distribution.

where $\sigma_0$ is the standard deviation considered for the distribution to be fitted to generate the downscaled data by the algorithm and $\sigma_{priordistribution}$ is the standard deviation of the obtained prior distribution. Later on, in section titled "Comparison of the Synthesized Data with the Real Data", we will see that the initial assumption of the standard deviation considered in (1) is a good approximation for the downscaled data.

**Initial data generation.** The *"Initial Data Generator"* phase feeds on the aggregated data, length of the aggregate interval, and $\sigma_0$ to give an initial downscaled data based on a "Distribution Generator". Based on the prior distribution, a proper statistical probability distribution (PD) is to be considered to be fitted to generate the data. The "Distribution Generator" aims to fit the selected PD to the prior distribution based on the statistical properties obtained for the initial phase. The challenge in this scenario and every step of the algorithm is ensuring that the synthetic data produced in every step is non-negative integers, as we are dealing with epidemiological data. Thus specific measures have been deployed to tackle these challenges, which are:

- To ensure non negativity consider the transformation:

$$\hat{\mathbf{y}} = \mathbf{y} + min(|\mathbf{y}|)$$

- To ensure that the data points are integer irrespective of the selection of PD, we round off the data to the nearest integer and subtract one from randomly selected data points in

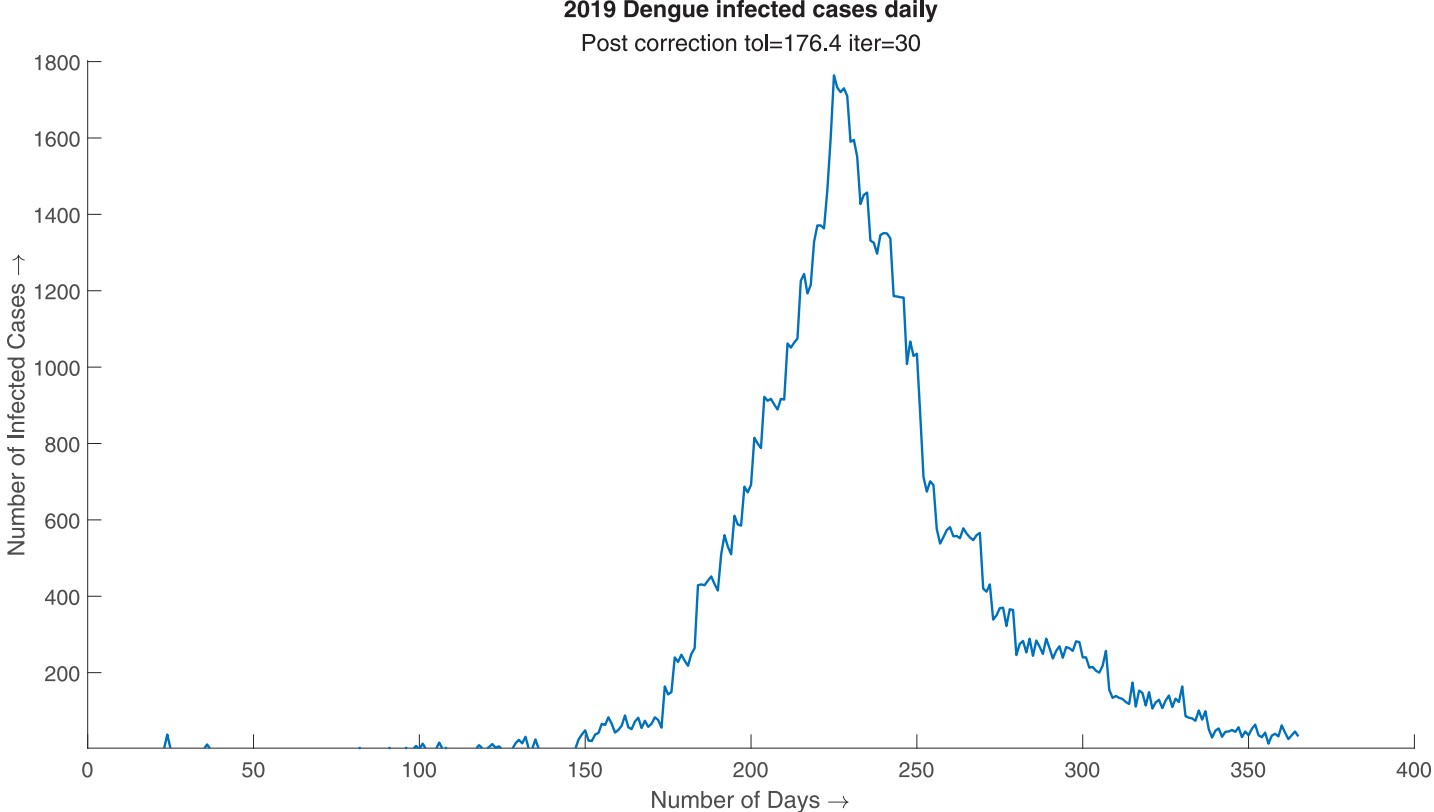

**Fig 5. 2019 Dengue infected cases daily, post correction tol = 176.4, iter = 30.** Initial approximation with overthrow correction exhibits a much proper approximation of the real case scenario preserving its original trend.

**Table 1. Comparison of actual vs SDB algorithm generated synthetic data using 2019 Dengue data of Bangladesh.**

| Month | Actual | Initial Distribution | Overthrow Correction | Volume Correction |
|---|---|---|---|---|
| January | 38 | 38 | 42 | 38 |
| February | 18 | 18 | 13 | 18 |
| March | 17 | 17 | 18 | 17 |
| April | 58 | 58 | 61 | 58 |
| May | 193 | 193 | 300 | 193 |
| June | 1884 | 1884 | 2500 | 1884 |
| July | 16253 | 16253 | 17617 | 16253 |
| August | 53636 | 53636 | 49581 | 53636 |
| September | 16856 | 16856 | 18259 | 16856 |
| October | 8143 | 8143 | 8419 | 8143 |
| November | 4011 | 4011 | 4094 | 4011 |
| December | 1247 | 1247 | 1450 | 1247 |
| **Total** | **102354** | **102354** | **102354** | **102354** |

The table exhibits the comparison of the number of cases each month for executing the SBD algorithm on the Dengue 2019 data of Bangladesh with the actual data. Here we can see the total number of infected individuals in each algorithm step is the same. In the case of the monthly sum, we see some anomaly in the overthrow correction case, which has been fixed in the volume correction step.

each aggregated unit such that the synthesized data has the same sum as the aggregated unit

Thus imposing these measures, the "Distribution Generator" generates synthetic distribution for each aggregated unit. Thus, looping over the entire aggregated timeline generates the initial distribution of the downscaled data concerning the aggregated data. This initial distribution is a suitable approximation to the actual data but can be improved with further refinement. The synthetic data will result in the exact aggregated data from which it is generated.

**Overthrow correction.** This step is often necessary for time series data with an abrupt change in gradient or in case of initial approximation with abnormally large overthrow as the approximations are probabilistic. In case of data with the abrupt change in gradient, the initial approximation is often left with a staircase-like structure as exhibited in the Fig 4. The problem can be corrected using the overthrow correction measure, which is demonstrated in Fig 5.

The overthrow correction part takes a tolerance, $\delta$, iteration limit, n, and a radius of an open interval, r. The step initially determines overthrow using tolerance between two neighboring points, i.e., if $y_i - y_{i-1} > \delta$ or $y_i - y_{i+1} > \delta$ then $y_i$ is an overthrow. After identifying an overthrow, we consider an open interval of radius r around the overthrow point and execute the distribution generator on that open interval. This redistributes the sample within the open interval diminishing the overthrow to some extent. This process is iterated n times over the entire time series to ensure satisfactory results. The strength of the overthrow correction step can be dictated by the two parameters $\delta$ and n. The strength of the overthrow correction is directly proportional to n and is inversely proportional to $\delta$. Selecting the correct parameter value can ensure a good approximation of the real-life scenario.

**Volume correction.** The overthrow correction disrupts the property of the synthesized time series to conserve its aggregated sum equal to the given aggregated distribution due to its local correction property. The scenario best illustrates the Table 1. This problem is addressed in this step. To maintain aggregated sum equal to the original data, we consider each aggregated unit and adjust the sum accordingly, adding/subtracting 1 from randomly chosen indices until the sum equates as required.

**The Stochastic Bayesian Downscaling (SBD) algorithm.** The algorithm calls for a unique name. From now on, we shall address it as Stochastic Bayesian Downscaling (SBD) algorithm. The structural part of the algorithm has been discussed at length in the first three segments of the methodology sub-section. The proper pseudo code of the SBD algorithm is as follows:

**Algorithm 1**. Stochastic Bayesian Downscaling (SBD) Algorithm

```
Require: Aggregated value vector, v
  Overthrow tolerance, δ
  Iteration limit, n
  Radius of the open interval, r
  Standard deviation, σ
Ensure: downscaled time series, v̄
  for elem in v do
    v̄ = Distribution Generator(elem,σ)
  end for
  for i from 1 to n do
    find a vector of coordinates of overthrow points
    for elem in overthrow points do
      open interval centering elem of radius, r = Distribution Genera-
tor(sum of the elements of open interval,σ)
    end for
  end for
  for elem in v do
```

```
if v_i ≠ sum of euiquivalent aggregate in v̄ then
  d=v_i-sum of equivalent aggregate in v̄
  while d ≠ 0 do
    if d > 0 then
      v̄_randomly picked index = v̄_randomly picked index + 1
      d− = 1
    else
      v̄_randomly picked index = v̄_randomly picked index − 1
      d− = 1
    end if
  end while
end if
end for
```

**Algorithm 2**. Distribution Generator

```
Require: Total sum of the down scaled distribution, s
  Standard deviation, σ
Ensure: Down scaled approximation over the length of the aggregate, v̄
  v̄ = Fit the decided distrubiton to the given down scaled time frame
  if elems in v̄ < 0 then
    v̄ = v̄ + |min(v̄)|
  end if
  if elems in v̄ are not integer then
    v̄ = round(v̄)
  end if
  if s ≠ ∑ v̄ then
    d = s − ∑ v̄
    while d ≠ 0 do
      if d > 0 then
        v̄_randomly picked index = v̄_randomly picked index + 1
        d− = 1
      else
        v̄_randomly picked index = v̄_randomly picked index − 1
        d− = 1
      end if
    end while
  end if
```

The SBD algorithm is heavily dependent on the random selection of numbers that are prone to generate non-reproducible results. Thus seeding the random number generator is highly recommended to ensure reproducible results.

The novelty of SBD algorithm is its consideration of the prior distribution as initialization and deploying the underlying distribution to generate synthesized downscaled data, which is non-negative and conserves the aggregated value of the given data.

## Comparison of the synthesized data with the real data

To determine the accuracy of the SBD algorithm, we test the SBD algorithm against some real-world data. Here, we have taken 2020 COVID-19 data on infected individuals in Bangladesh and 2022 (January to July), Dengue data on infected individuals in Bangladesh. The data, as mentioned earlier, are daily data on the number of newly infected individuals nationwide. We aim to convert this data to monthly aggregate and feed the aggregated data to the algorithm to generate downscaled daily data; hence we can compare the accuracy of the synthetic daily data with the actual daily data. To determine the accuracy of the approximation, we will use two error measures and do component analysis on the real and synthetic data to see if the synthetic data can well approximate the underlying properties of the real data. In case of the component

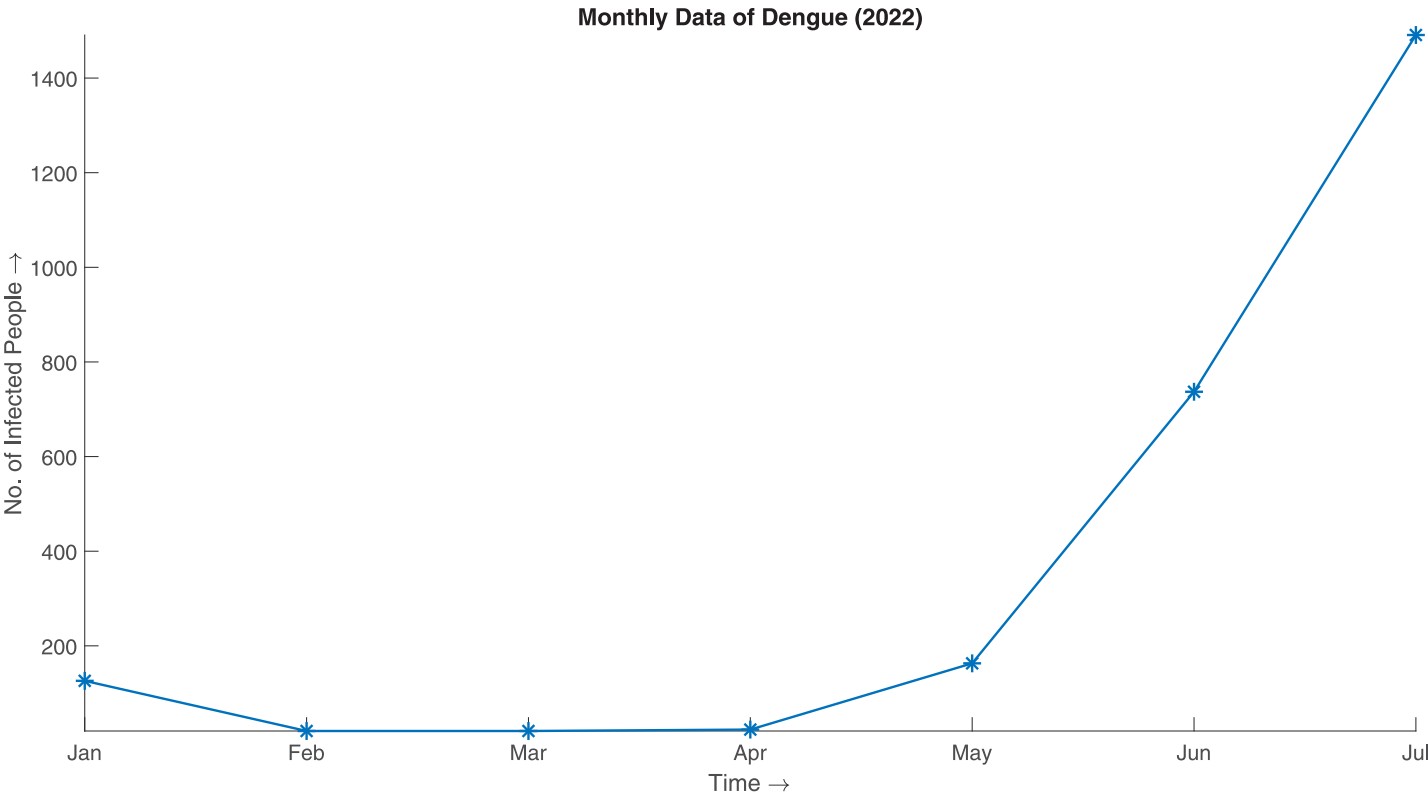

**Fig 6. Monthly data of Dengue (2022).** Monthly aggregate of 2022 Dengue data from January to July.

decomposition, we will use the additive model mentioned in (2),

$$y_i = Trend + Seasonality + Residual \qquad (2)$$

as the procured data has some zero values for which the multiplicative model mentioned in (3)

$$y_i = Trend \times Seasonality \times Residual \qquad (3)$$

is not suitable in this scenario.

### Error measures for benchmark

To compare the result with the real world data we shall use two error terms that describes the overall error of the approximation. These are as follows:

- **Root Mean Square Error**:

$$\text{RMSE} = \sqrt{\frac{\sum_{i=1}^{N}(x_i - \hat{x}_i)}{N}}$$

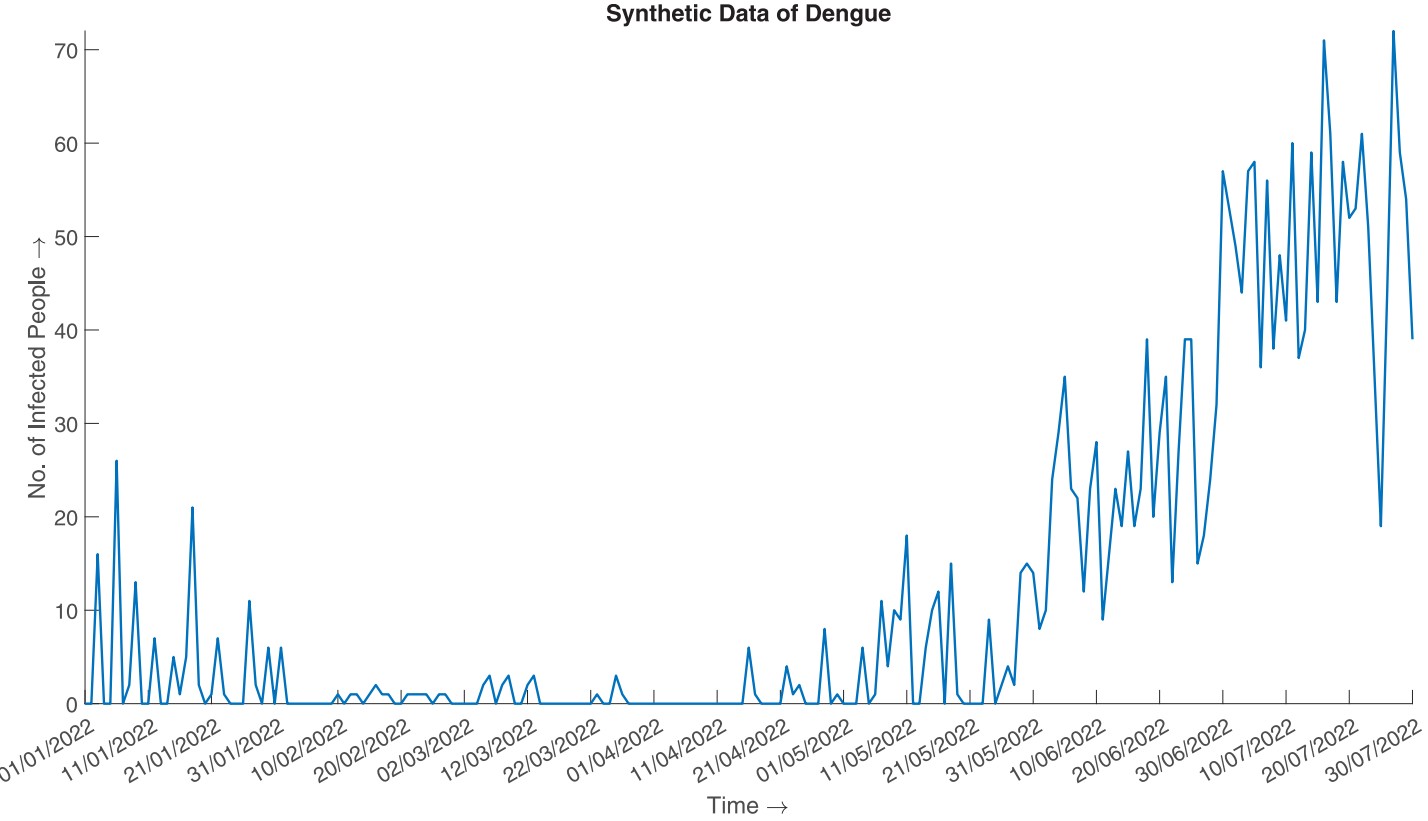

**Fig 7. Synthetic data of Dengue.** SDB algorithm generated synthesized daily number of infected cases of Dengue in 2022 from January to July.

- **Mass Absolute Error**

$$\text{MAE} = \frac{\sum_{i=1}^{N} |x_i - \hat{x}_i|}{N}$$

where, $x_i$ is the actual data and $\hat{x}_i$ is the predicted data.

Since many of the data points in the actual and synthesized cases is popluated with 0 hence Mass Absolute Percentage Error (MAPE), and Scaled Mass Absolute Percentage Error (SMAPE) are undefined in this scenario.

## Dengue

**Preprocessing and result.** In case of this simulation, we took Bangladesh's 2022 daily Dengue infected data from January to July. To feed this data into the SBD algorithm, we convert the daily data to monthly aggregate as illustrated in Fig 6. For majority of the statistical work done in the paper we have used R.

We feed in this data considering,

- Initial standard deviation, $\sigma_0 = \dfrac{\sigma_{prior\ distribution}}{30} = \dfrac{556.6431703}{30} = 18.55477234$.

- Over throw tolerance, $\delta = 0.6\times$ (Range of the initial distribution).

- Iteration limit, $n = 100$.

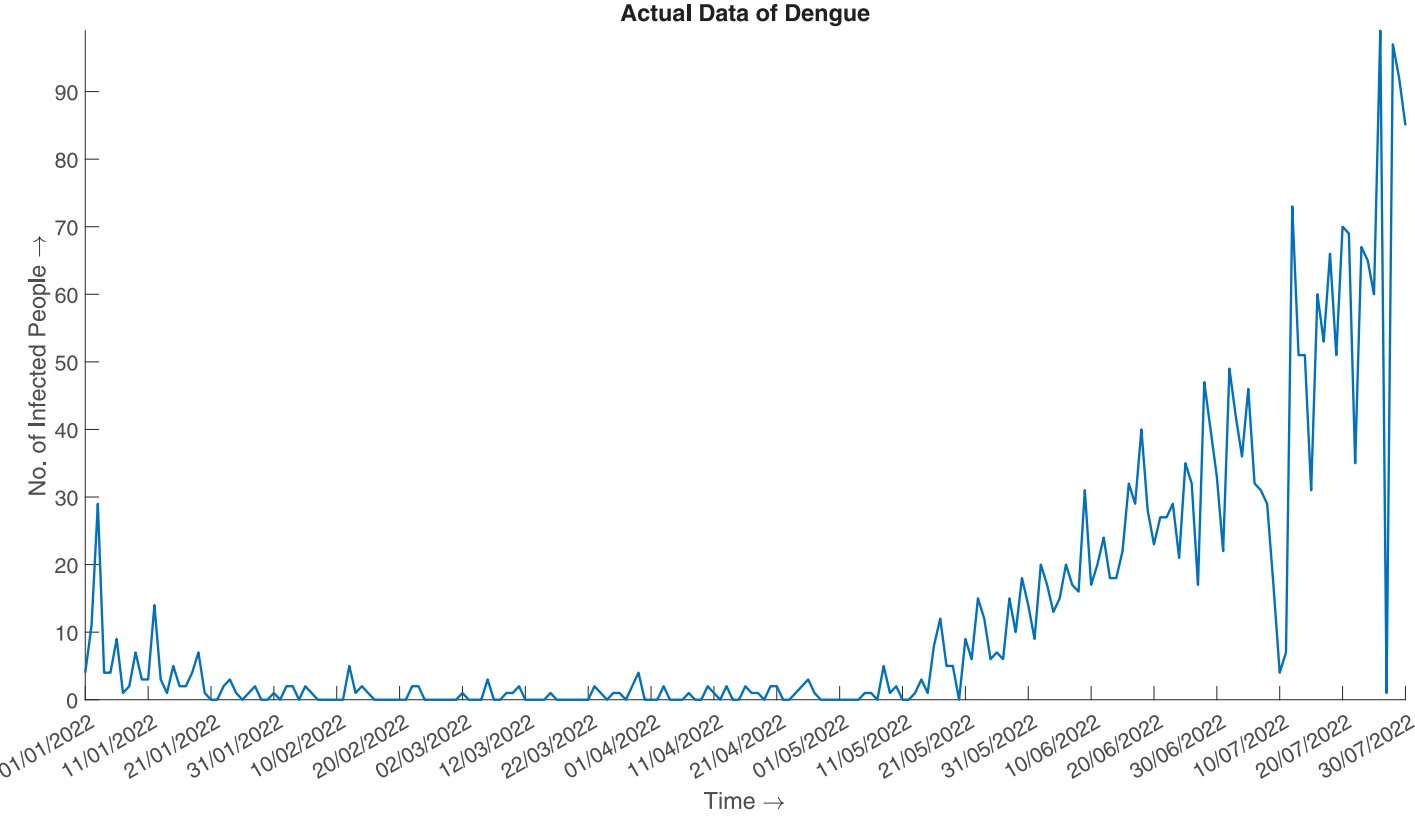

**Fig 8. Actual data of Dengue.** Daily number of infected cases of Dengue in 2022 from January to July.

- Radius of open interval, $r = 3$.

- Underlying distribution to be normal.

and generate the synthesized data. Fig 7 illustrates the synthesized data, which can be said to be a good approximation of the actual given the aggregated prior distribution (Fig 8).

**Table 2. Aggregation comparison of actual vs. SBD algorithm generated synthetic data in the case study of Dengue.**

| Month | Actual | Initial Distribution | Overthrow Correction | Volume Correction |
|---|---|---|---|---|
| January | 126 | 126 | 119 | 126 |
| February | 20 | 20 | 27 | 20 |
| March | 20 | 20 | 20 | 20 |
| April | 23 | 23 | 32 | 23 |
| May | 163 | 163 | 206 | 163 |
| June | 737 | 737 | 733 | 737 |
| July | 1491 | 1491 | 1443 | 1491 |
| **Total** | **2580** | **2580** | **2580** | **2580** |

The column named "Actual" represent the actual data and the following three columns (mentioned in its order of progression) represent the aggregation of downscaled data (generated for the purpose of comparison and validation) at each phase of SBD algorithm. The final outcome of the algorithm denoted in the table as "Volume Correction" column is in agreement with the actual data. A distinction of this algorithm is that at each phase of the process, the SDB algorithm maintains the total sum of the over aggregation unit (in this case month) equal to that of the actual data which helps to maintain the accuracy of the synthetic data.

**Table 3. Statistical measure comparison of 2022 actual Dengue data vs. SDB algorithm generated data.**

| Measures | Observed | Synthesized |
|---|---|---|
| Mean | 12.22748815 | 12.22748815 |
| Standard Deviation | 20.28993189 | 18.49672823 |
| Minimum | 0 | 0 |
| Lower Quartile(Q1) | 0 | 0 |
| Median | 2 | 1 |
| Upper Quartile(Q2) | 17 | 19 |
| Maximum | 99 | 72 |

This table illustrates the comparison of the basic statistical measures of the synthesized data with respect to the actual data. The second and third column represent the statistical measures observed for the actual data and algorithm generated data respectively. The synthetic data was able to replicate the mean of the actual data exactly with out being provided with it. The rest of the statistics are close enough approximation except that of the maximum value. A smart way to achieve this is can be an open avenue for research.

**Error metrics and statistical measures.** The calculated error measures are:

- $MAE$ = 6.60664, which implies that the average error between the actual and synthesized data is 6.60664.

- $RMSE$ = 12.64499 which implies that the standard deviation of the residuals/errors is 12.64499. The fact is well illustrated in Fig 14.

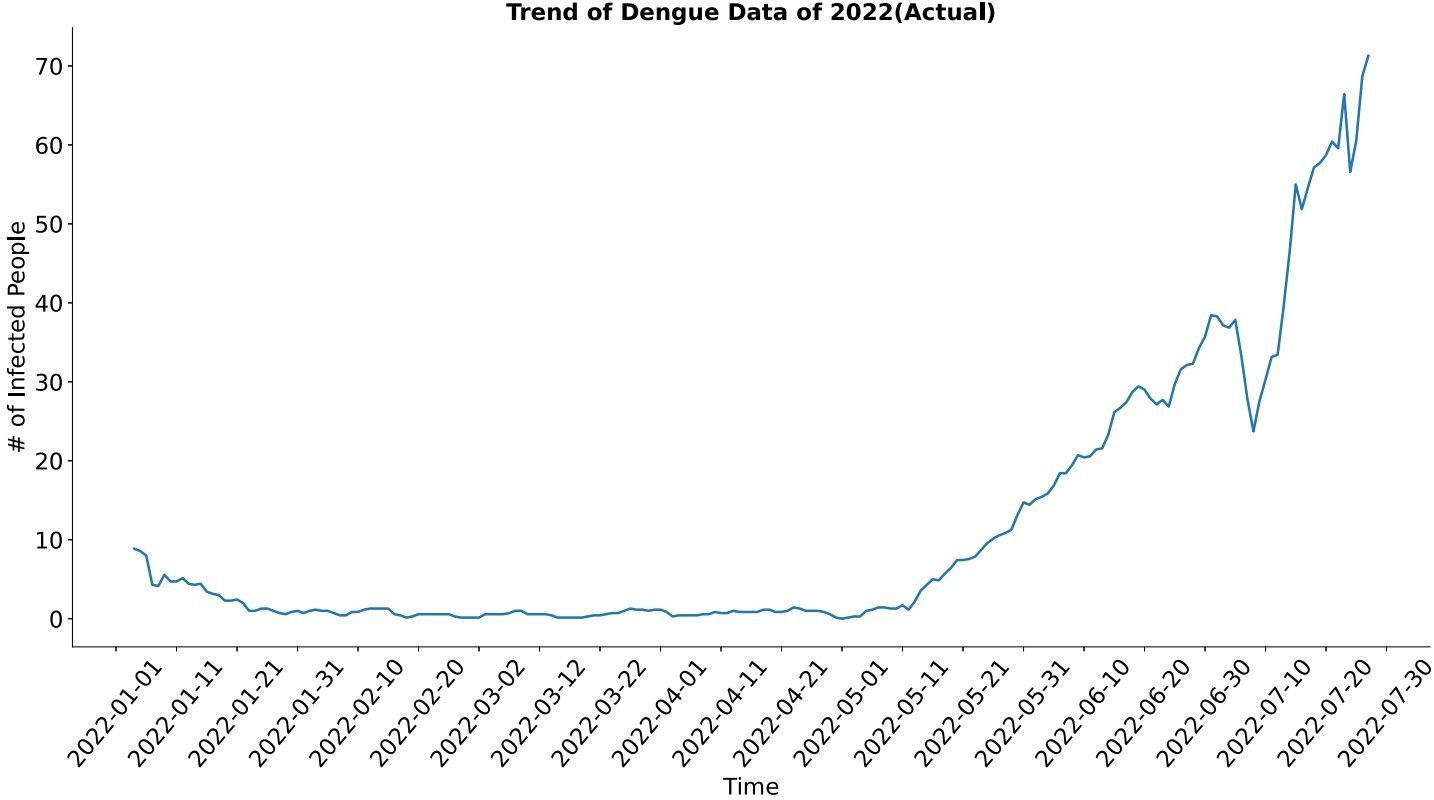

**Fig 9. Trend of Dengue data of 2022 (Actual).** Trend of the actual dengue data.

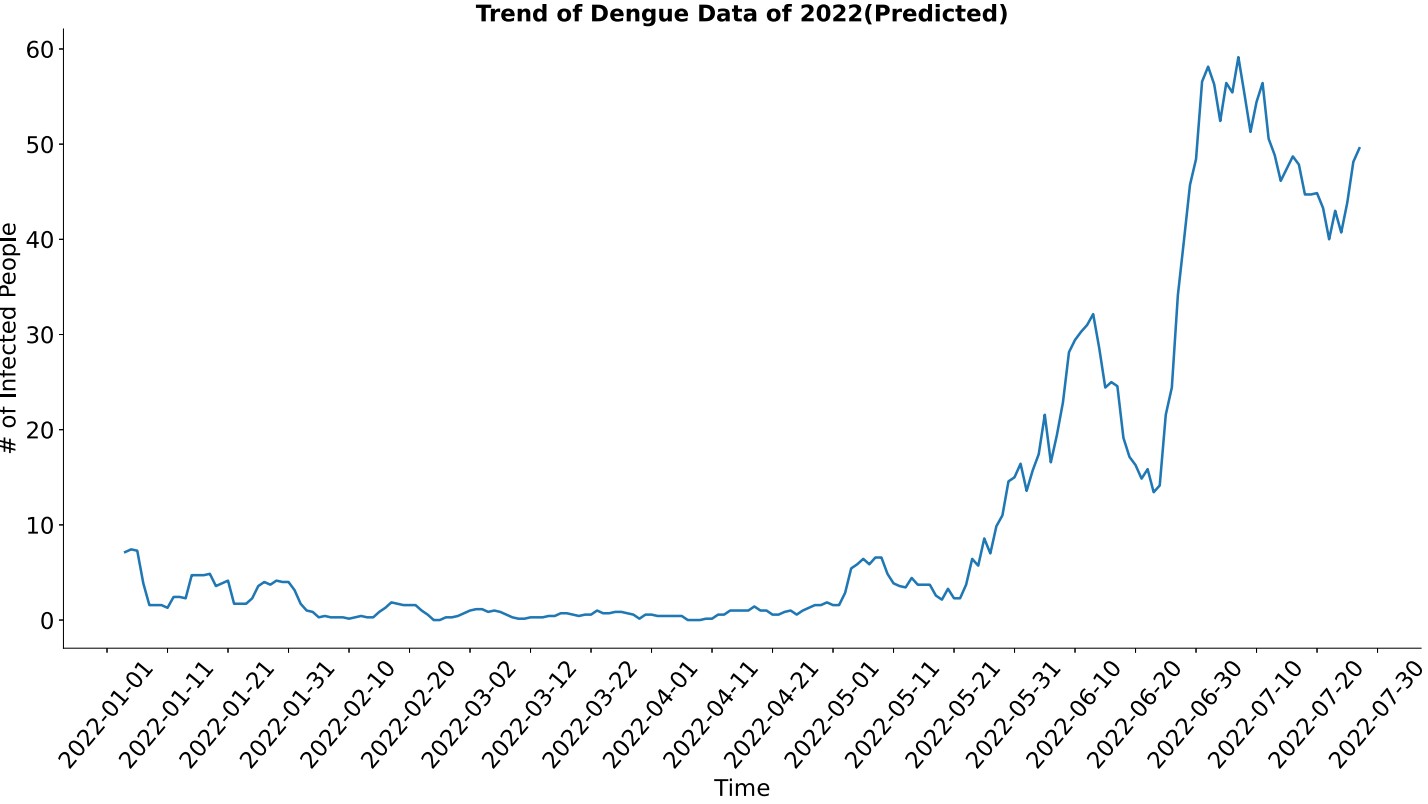

**Fig 10. Trend of Dengue data of 2022 (Predicted).** Trend of the synthetic dengue data.

The error metric shows satisfactory results. The following Table 2 validates if the synthesized data honours the aggregated sum of the prior distribution.

The total number of cases in each scenario has been maintained equally. As discussed earlier, we can see that the initial distribution holds the monthly sum consistently, which gets disrupted in the overthrow correction phase and later corrected in the volume correction phase.

We shall now explore the basic statistical properties of the synthetic data with respect to the actual data.

It is to be noted that the mean of the synthesized data equates to that of the original data, although it was not plugged into the SBD algorithm in any manner as illustrated in Table 3. As previously discussed that $\sigma_0$ is a good approximation to the original $\sigma$. All the rest of the measures are somewhat close, but the maximum varies by a lot. The maximum is hard to anticipate from the aggregated data; hence it is an avenue that demands further exploration.

**Component decomposition and comparison.**   We now want to do component decomposition of both the actual and synthetic data based on the model mentioned in (2). However, component decomposition is no benchmark for accuracy, but SBD algorithm aims to improve the outcome of forecasting techniques highly influenced by the components within a time series data. Thus comparing these components can answer the question of whether the components-based characteristics of the original time series are present within the synthesized data.

In the case of the trend component (Figs 9 and 10), both the actual and the synthesized data shows similar result and trend of the actual data have been well approximated by the trend of the synthesized data.

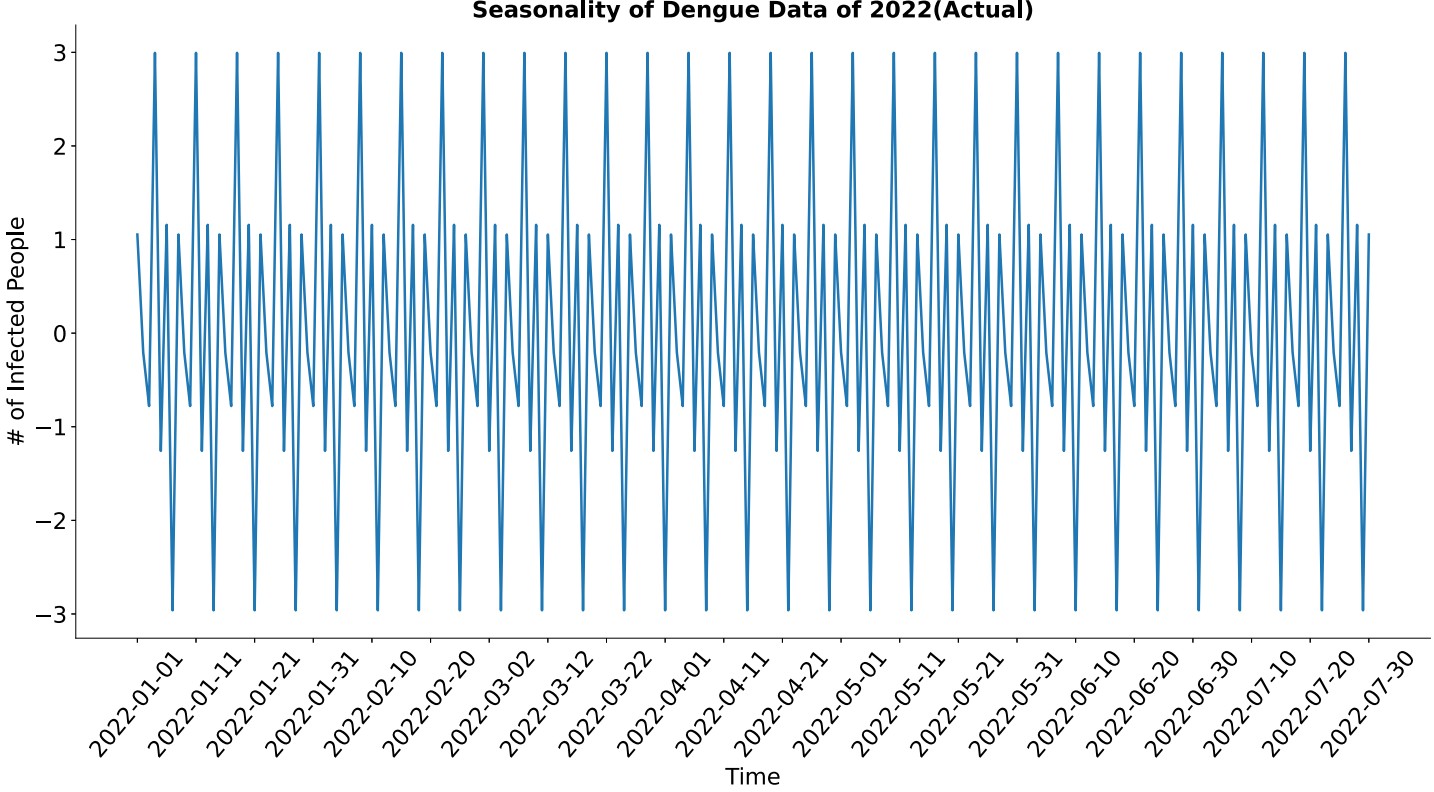

**Fig 11. Seasonality of Dengue data of 2022 (Actual).** Seasonality of the actual dengue data.

In the case of the seasonality component (Figs 11 and 12), both the actual and the synthesized data show major weekly and minor sub-weekly seasonality. The synthesized data's seasonality approximates the actual data's seasonality well.

In the case of the residual component (Figs 13 and 14), both the actual and the synthesized data show a similar result, although the residual of the synthetic data may look noisy at first glance but upon closer inspection, it is evident that the residual of the synthetic data shows less deviation from the standard value in comparison to the actual data. The synthesized data's residual has well approximated the actual data's residual.

As mentioned earlier, the key takeaway from the discussion is that the SBD algorithm could generate an excellent approximation of the dengue data from the monthly aggregated data based on some statistical properties of the prior distribution. In the following section, we shall also test SBD algorithm's efficacy in another epidemiological scenario.

## COVID-19

**Preprocessing and result.** In case of this simulation, we took Bangladesh's 2020 daily COVID-19 infected data from March to December [29, 30]. To feed this data into the SBD algorithm, we convert the daily data to monthly aggregate as illustrated in Fig 15,

We feed in this data considering,

- Initial standard deviation, $\sigma_0 = \dfrac{\sigma_{prior\ distribution}}{30} = \dfrac{32021.87439}{30} = 1067.395813$.

- Over throw tolerance, $\delta = 0.2 \times$ (Range of the initial distribution).

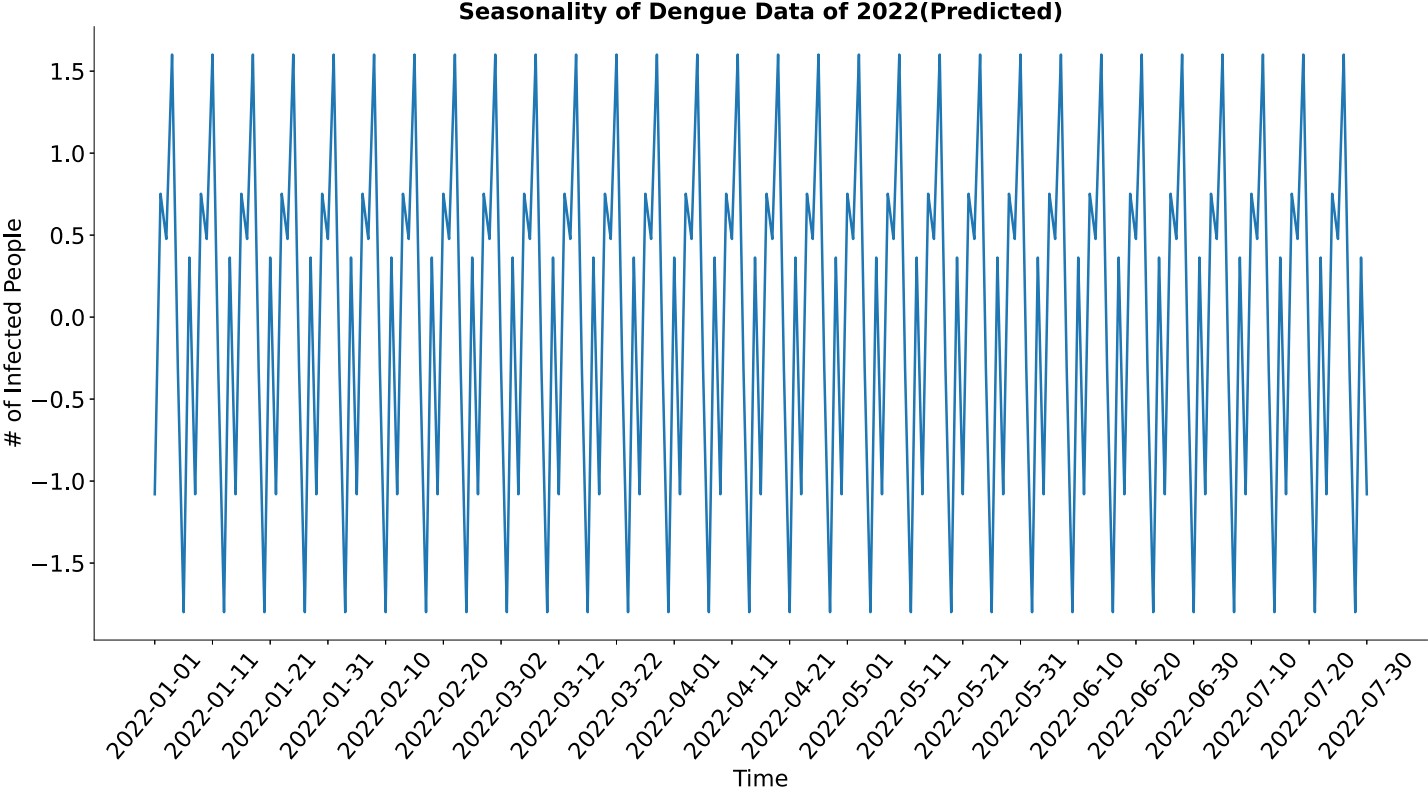

**Fig 12. Seasonality of Dengue data Of 2022(Predicted).** Seasonality of the synthetic dengue data.

- Iteration limit, $n = 100$.

- Radius of open interval, $r = 3$.

- Underlying distribution to be normal.

and generate the synthesized data. Fig 16 illustrates the synthesized data, which can be said to be a good approximation of the actual given the aggregated prior distribution (Fig 17).

**Error metrics and statistical measures.** The calculated error measures are:

- *MAE* = 257.41806, which implies that the average error between the actual and synthesized data is 257.41806, which is reasonable considering the mean of the data is 1717.424749.

- *RMSE* = 346.6241, which implies that the standard deviation of the residuals/errors is 346.6241. The fact is well illustrated in Fig 23.

it is to be noted that the error term of this scenario must not be compared with the error term of the previous case as they are of varying scale. Compared to the scale of the data, the error metric shows satisfactory results. The following Table 4 validates if the synthesized data honours the aggregated sum of the prior distribution.

We shall now explore the basic statistical properties of the synthetic data with respect to the actual data.

It is to be noted that the mean of the synthesized data equates to that of the original data, although it was not plugged into the SBD algorithm in any manner as illustrated in Table 5. As previously discussed that $\sigma_0$ is a good approximation to the original $\sigma$. All the rest of the

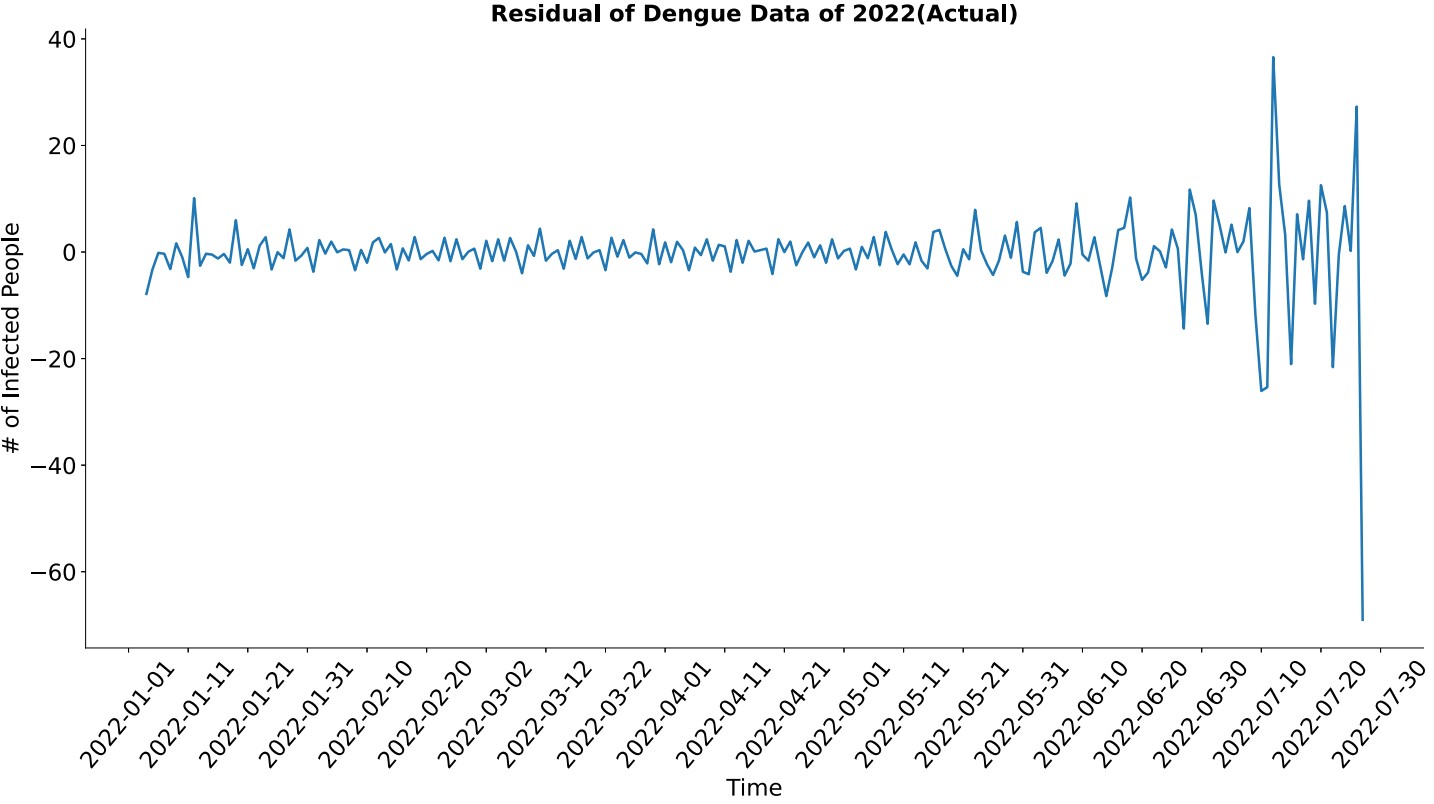

**Fig 13. Residual of Dengue data of 2022(Actual).** Residual of the actual dengue data.

measures are somewhat close, but the maximum varies by a lot. The maximum is hard to anticipate from the aggregated data; hence it is an avenue that demands further exploration.

**Component decomposition and comparison.** We now want to do component decomposition of both the actual and synthetic data based on the model mentioned in (2). However, component decomposition in no way is a benchmark for accuracy, but as SBD algorithm aims to improve the outcome of forecasting techniques which are highly influenced by the components within a time series data. Thus, comparing these components can answer the question of whether the original time series's components-based characteristics are present in the synthesized data.

In case of the trend component (Figs 18 and 19) both the actual and the synthesized data shows similar result and trend of the actual data have been well approximated by the trend of the synthesized data.

In case of the seasonality component (Figs 20 and 21), both the actual and the synthesized data shows major weekly seasonality. The seasonality of the synthesized data has well approximated the seasonality of the actual data.

In the case of the residual component (Figs 22 and 23), both the actual and the synthesized data shows a similar result, although the residual of the synthetic data may look a bit noisy at first glance but upon closer inspection, it is evident that the residual of the synthetic data shows less deviation from the standard value in comparison to the actual data. The residual of the synthesized data has well approximated the residual of the actual data.

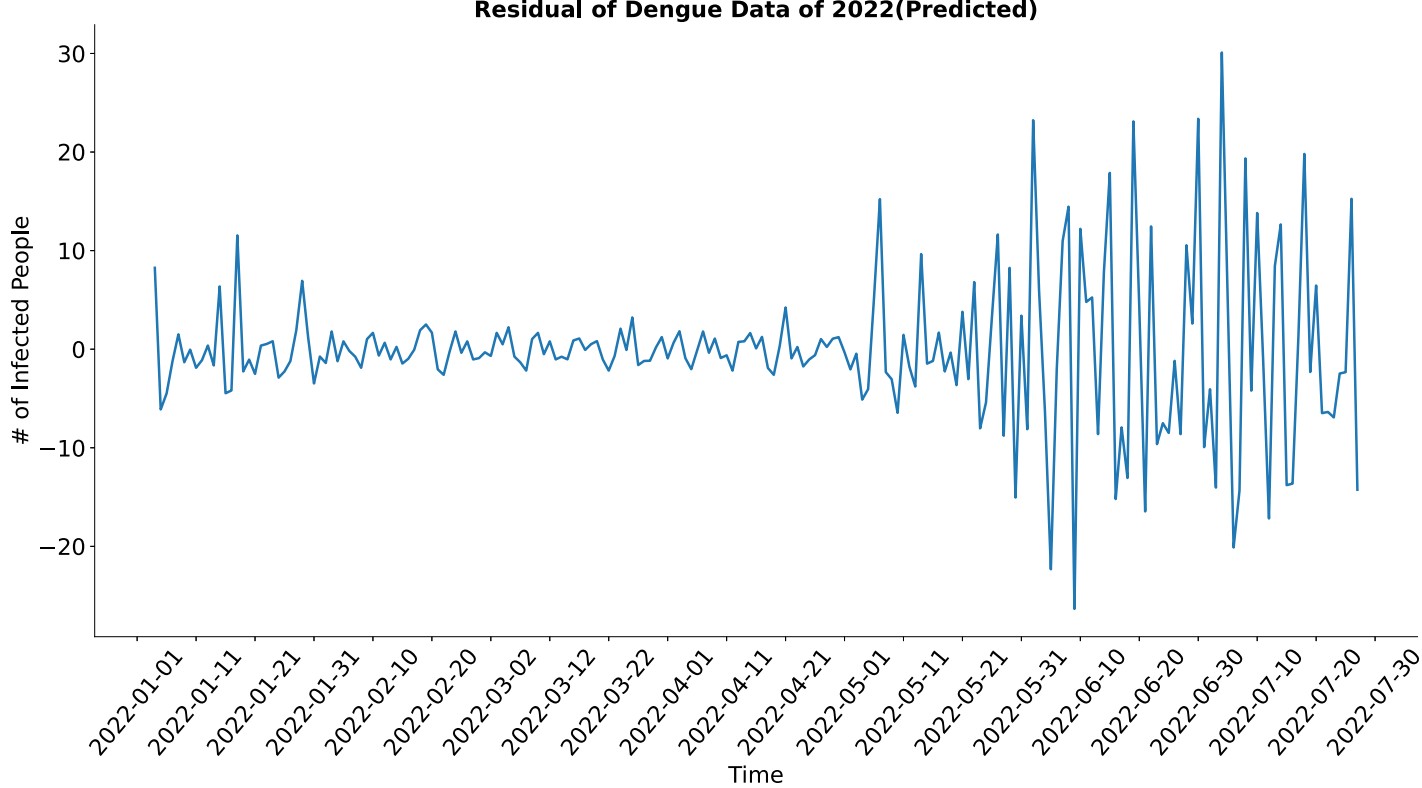

**Fig 14. Residual of Dengue data of 2022(Predicted).** Residual of the synthetic dengue data.

The key takeaway from the discussion above is that the algorithm could generate an excellent approximation of the COVID-19 data from the monthly aggregated data based on some statistical properties of the prior distribution. We shall also test SBD algorithm's efficacy in a forecasting scenario in the following section.

## Improvements in forecasting accuracy

In this section, we shall forecast the Dengue infection case in Bangladesh using statistical forecasting tools. The use of statistical modelling is one of the helpful ways that may be utilized for the forecasting of dengue outbreaks [31, 32]. Previous research carried out in China [33], India [34], Thailand [35], West Indies [36], Colombia [37], and Australia [38] on infectious diseases made substantial use of the time series technique in the field of epidemiologic research on infectious diseases [38]. A number of earlier research looked at the Autoregressive Integrated Moving Average (ARIMA) model as a potential tool for use in forecasting [39–44].In addition, the ARIMA models have seen widespread use for dengue forecasting [42, 45–47]. When establishing statistical forecasting models, these are frequently paired with Seasonal Auto-regressive Integrated Moving Average (SARIMA) models, which have proven to be suitable for assessing time series data with ordinary or seasonal patterns [34, 36, 38, 48–50]. It is likely that developing a dengue incidence forecasting model based on knowledge from previous outbreaks and environment variables might be an extremely helpful tool for anticipating the severity and frequency of potential epidemics.

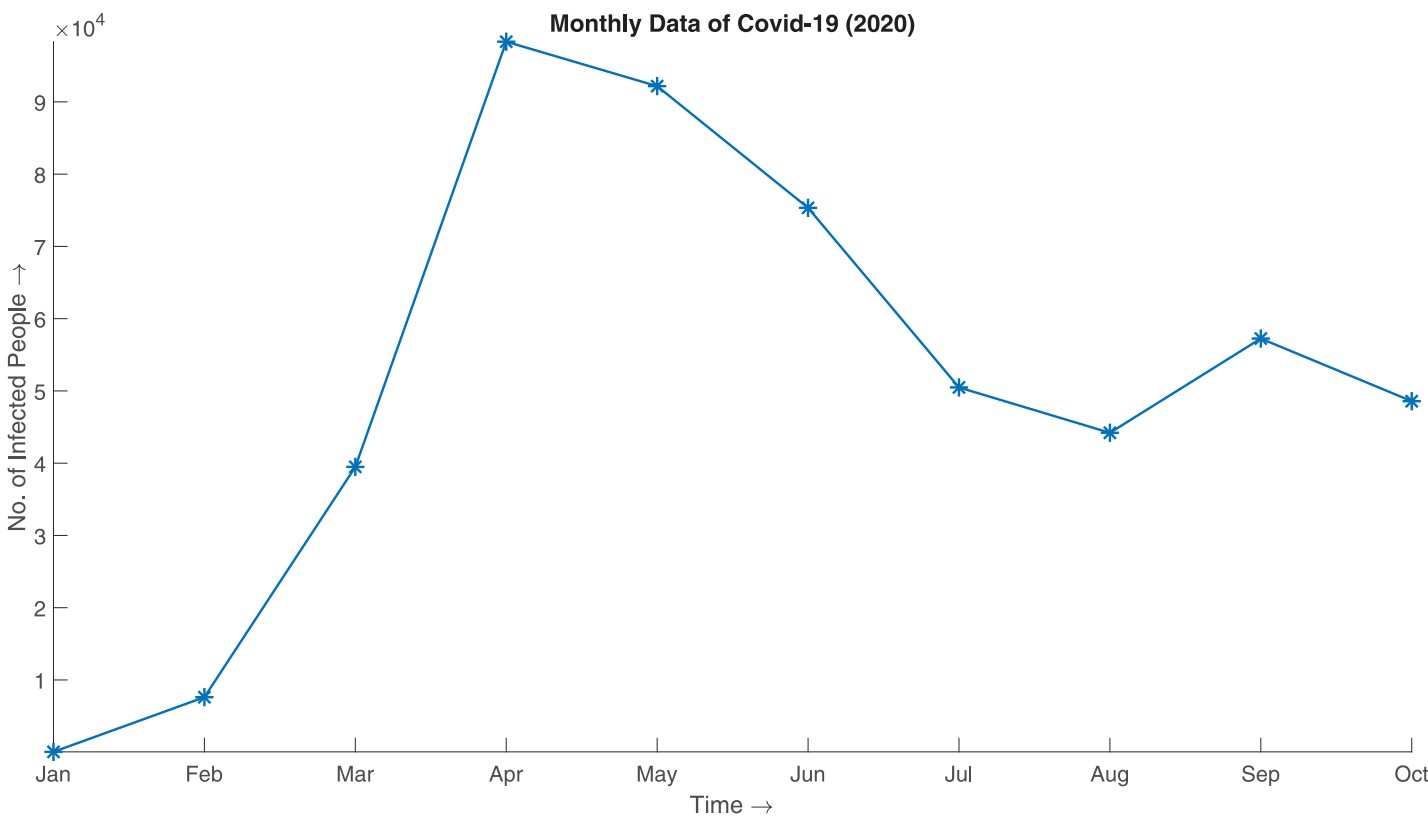

**Fig 15. Monthly data of Covid-19 (2020).** Monthly aggregate of 2020 COVID-19 infected data of Bangladesh from March to December.

The idea of seasonality using the Fourier coefficient naming Fourier ARIMA model was introduced by [51, 52].

$$Z_t = \delta_0 + \sum_{i=1}^{p} \alpha_i Z_{t-i} + \sum_{j=1}^{q} \beta_J e_{t-j} + \sum_{k=1}^{r} [a_k \sin(\omega_k t) + b_k \cos(\omega_k t)] Z_{t-m} + e_t \tag{4}$$

where, $\delta_0$ is the constant term and $\omega_k$ is the periodicity of the data.

We aim to forecast the monthly and synthesized daily data using the forecasting mentioned above techniques and compare the forecast accuracy based on error measures. We use SARIMA and Fourier-ARIMA models to forecast the monthly and synthesized data. The model in each case is chosen based on the lowest value of Akaike's Information Criterion (AIC), Akaike's Information Criterion correction (AICc), and Bayesian Information Criterion (BIC).

## Model selection method

Box-Jenkins method is a generalized model selection pathway which works for time series irrespective of its stationarity or seasonality. The method is illustrated in Fig 24.

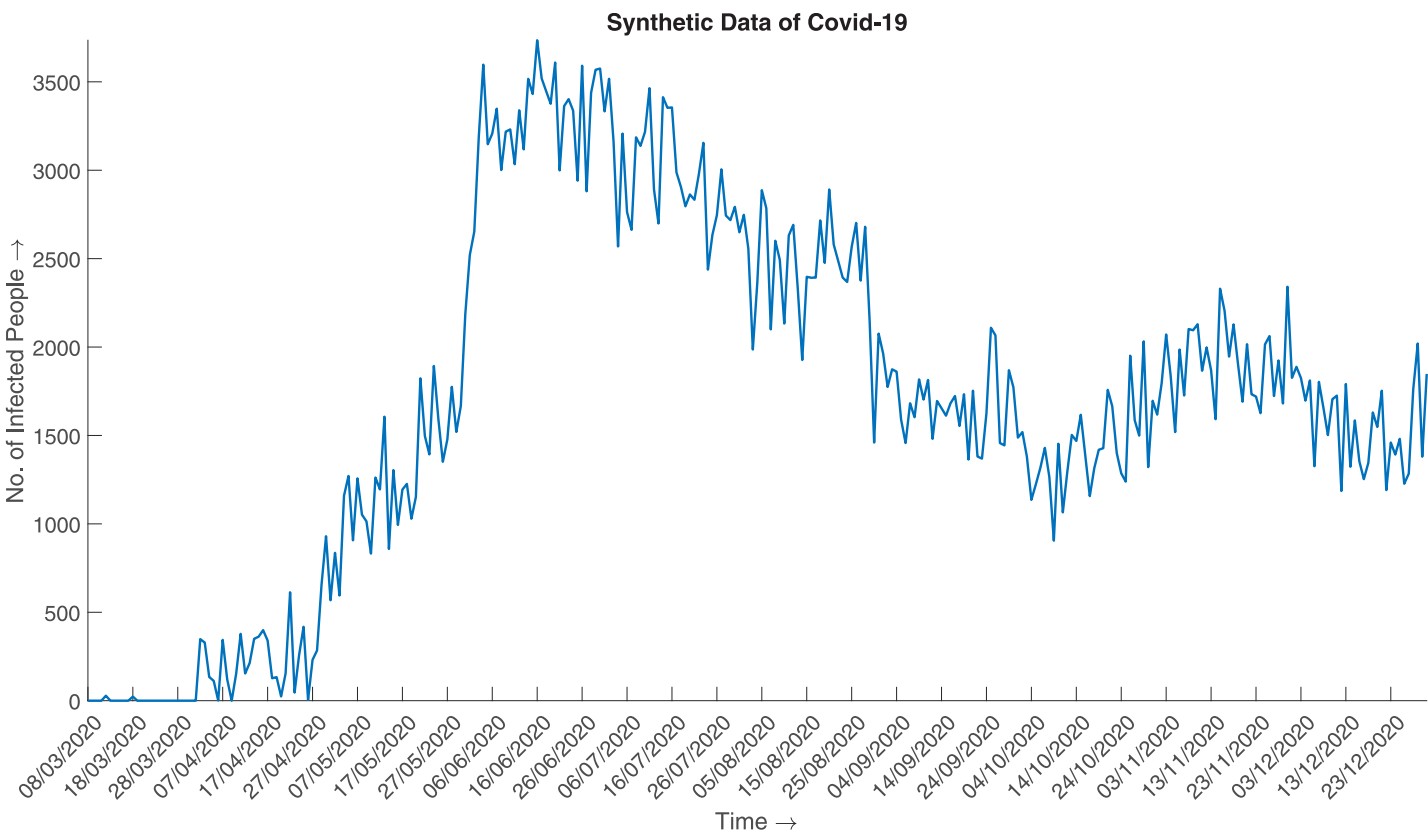

**Fig 16. Synthetic data of Covid-19.** Synthesized daily number of infected cases of COVID-19 in 2020 from March to December.

### Error measures of model

The error measures for comparison is Mean Absolute Scaled Error(MASE) which is defined as

$$MASE = \frac{\frac{1}{n}\sum_{i=1}^{n}|Y_i - \hat{Y}_i|}{\frac{1}{T-m}\sum_{t=m+1}^{T}|Y_t - Y_{t-m}|}$$

We used this metric as it is scale-independent; hence is perfect for comparison [53, 54]. We also could have taken MAPE as a metric, but MAPE is undefined for such cases as the data is populated with zero values. We also use RMSE and MAE to gauge the error in the forecast [55, 56].

### Forecast on the aggregated data

The actual data is monthly Dengue infection data of Bangladesh from 2010 to July 2022. Following Box-Jenkin's method, we firstly check for the stationarity of the data based on the Augmented Dicky Fuller (ADF) test. ADF test returns the value of -4.7906 with p-value = 0.01, which implies that the data is stationary.

We run multiple SARIMA models and calculate their AIC, AICc and BIC and the best model is chosen based on the minimum value of the criterion. We present 5 of the top results in Table 6.

Here, the best model to use is SARIMA $(1, 0, 0)(0, 1, 1)_{12}$. We fit the given model, which gives us the coefficients presented in Table 7:

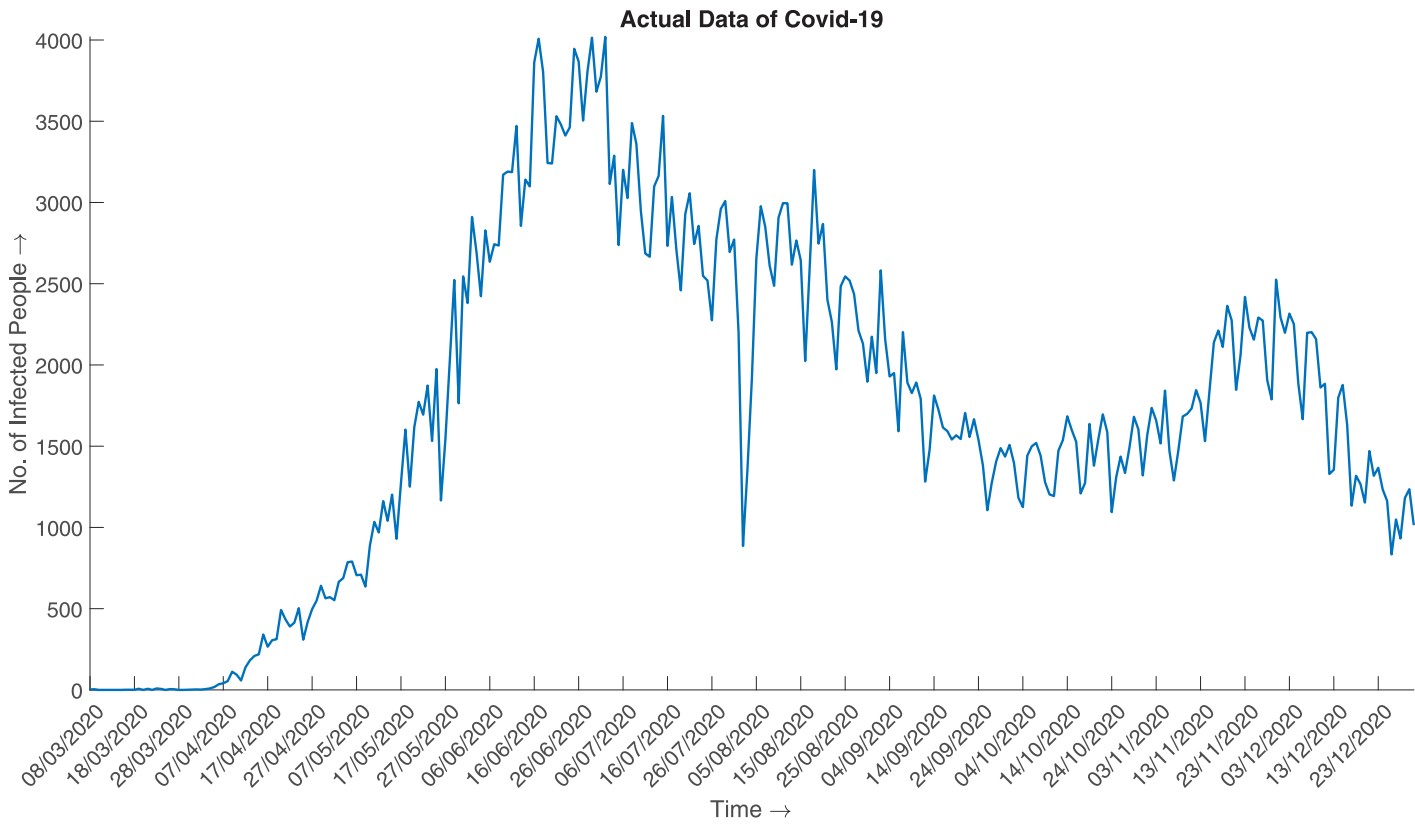

**Fig 17. Daily data of Covid-19 (2020).** Daily number of infected cases of COVID-19 in 2020 from March to December.

To check the goodness of fit of the model, we use the Ljung box test, which returns the p-value = 0.9996 > 0.05, i.e. we accept the null hypothesis: *"The model does not show lack ness of fit/ the residuals are not autocorrelated/ the residuals are random white noise."*

Given everything in place, we forecast the infection for the rest of 2023, i.e. from August to December. The forecast is illustrated in the given figure (Fig 25).

**Table 4. This table illustrates that the synthetic data agrees with the monthly sum of the actual data.**

| Month | Actual | Initial Distribution | Overthrow Correction | Volume Correction |
|---|---|---|---|---|
| March | 51 | 51 | 51 | 51 |
| April | 7616 | 7616 | 9226 | 7616 |
| May | 39486 | 39486 | 41261 | 39486 |
| June | 98330 | 98330 | 94075 | 98330 |
| July | 92178 | 92178 | 92115 | 92178 |
| August | 75335 | 75335 | 75605 | 75335 |
| September | 50483 | 50483 | 50766 | 50483 |
| October | 44205 | 44205 | 45126 | 44205 |
| November | 57248 | 57248 | 55805 | 57248 |
| December | 48578 | 48578 | 49480 | 48578 |
| **Total** | **513510** | **513510** | **513510** | **513510** |

The total number of cases in each scenario has been maintained equally. As discussed earlier, we can see that the initial distribution holds the monthly sum consistently, which gets a little disrupted in the overthrow correction phase and is later on corrected in the volume correction phase.

**Table 5. Statistical measure comparison of actual vs. SDB algorithm generated synthetic data.**

| Measures | Observed | Synthesized |
|---|---|---|
| Count | 299 | 299 |
| Mean | 1717.424749 | 1717.424749 |
| Standard Deviation | 1044.457258 | 1007.554237 |
| Minimum | 0 | 0 |
| Lower Quartile(Q1) | 1115.5 | 1225 |
| Median | 1666 | 1696 |
| Upper Quartile(Q2) | 2521.5 | 2481.5 |
| Maximum | 4019 | 3735 |

This table illustrates the comparison of the basic statistical measures of the synthesized data with respect to the actual data. The second and third column represent the statistical measures observed for the actual data and algorithm generated data respectively. The synthetic data was able to replicate the mean of the actual data exactly with out being provided with it. The rest of the statistics are close enough approximation except that of the maximum value. A smart way to achieve this is can be an open avenue for research.

To validate the goodness of the fit, we can analyze the model residual, illustrated in Fig 26. Here, the top graph is the residual with the timeline of the original data. The bottom left graph represents the Autocorrelation Function (ACF) with respect to the lag of the data. Almost all the values are within the significance e level, and the bottom right figure shows the distribution of the model's residuals. It implies that the residuals are distributed generally with zero means.

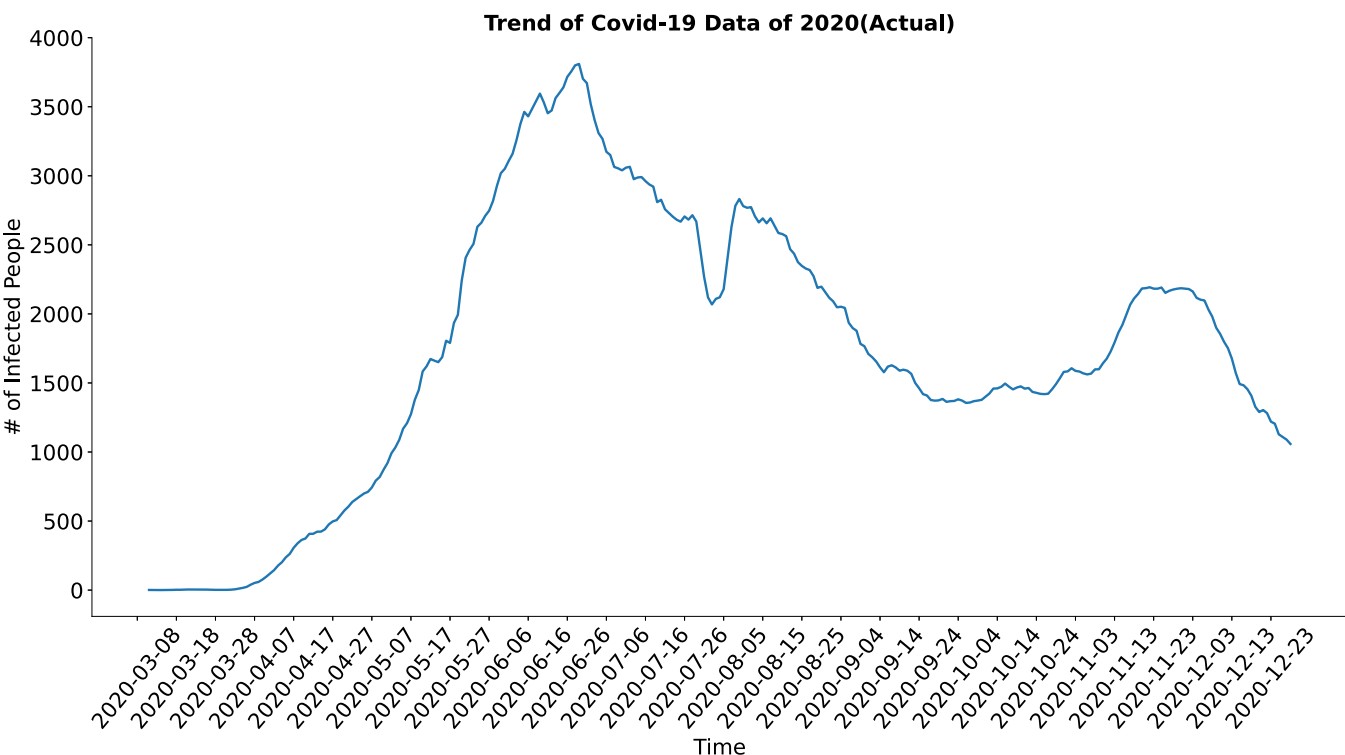

**Fig 18. Trend of Covid-19 data of 2020(Actual).** Trend of the actual COVID-19 data.

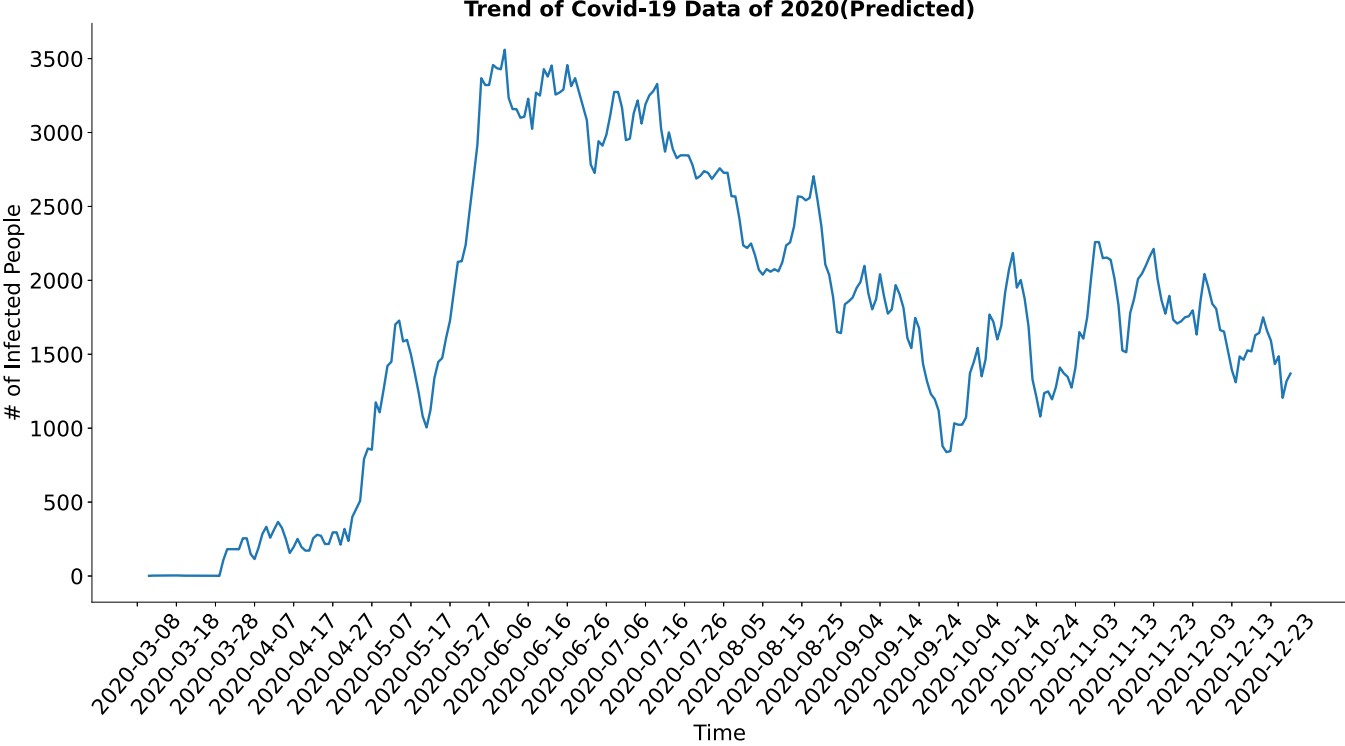

**Fig 19. Trend of Covid-19 data of 2020(Predicted).** Trend of the synthetic COVID-19 data.

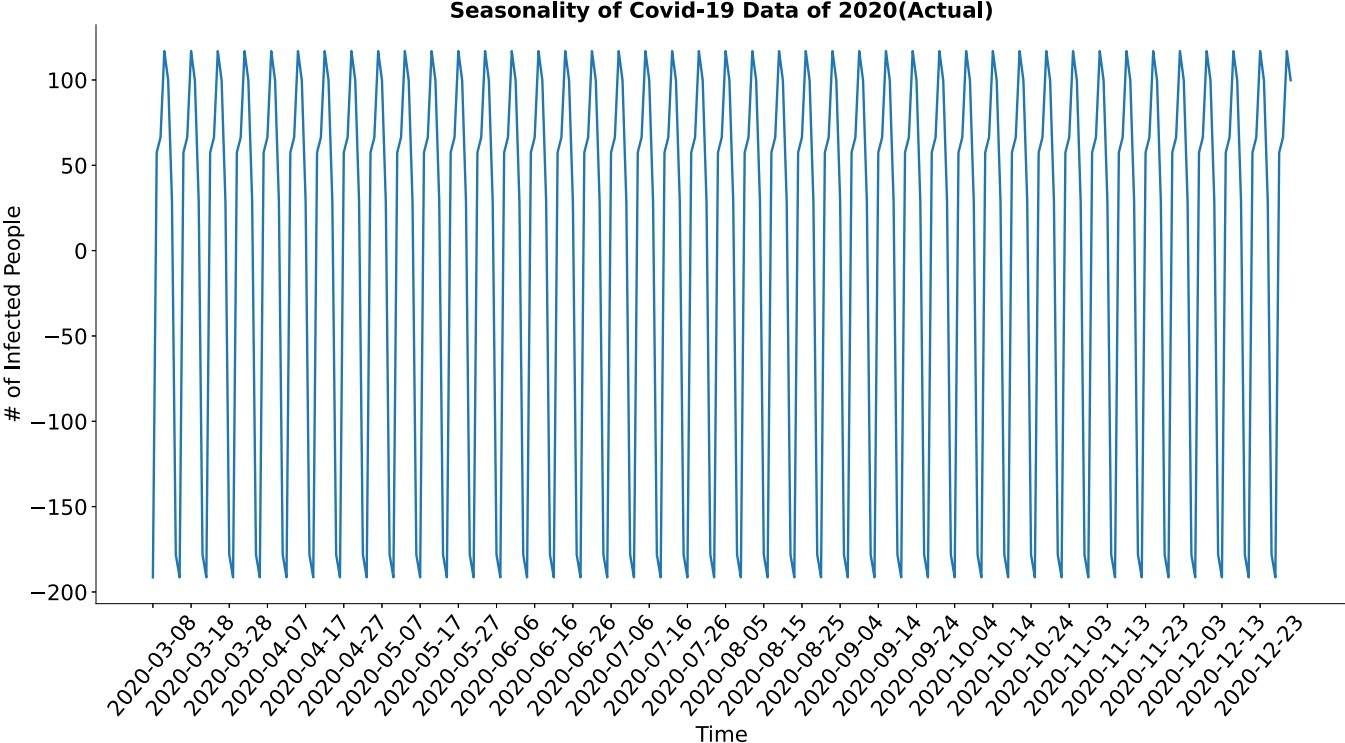

**Fig 20. Seasonality of Covid-19 data of 2020(Actual).** Seasonality of the actual COVID-19 data.

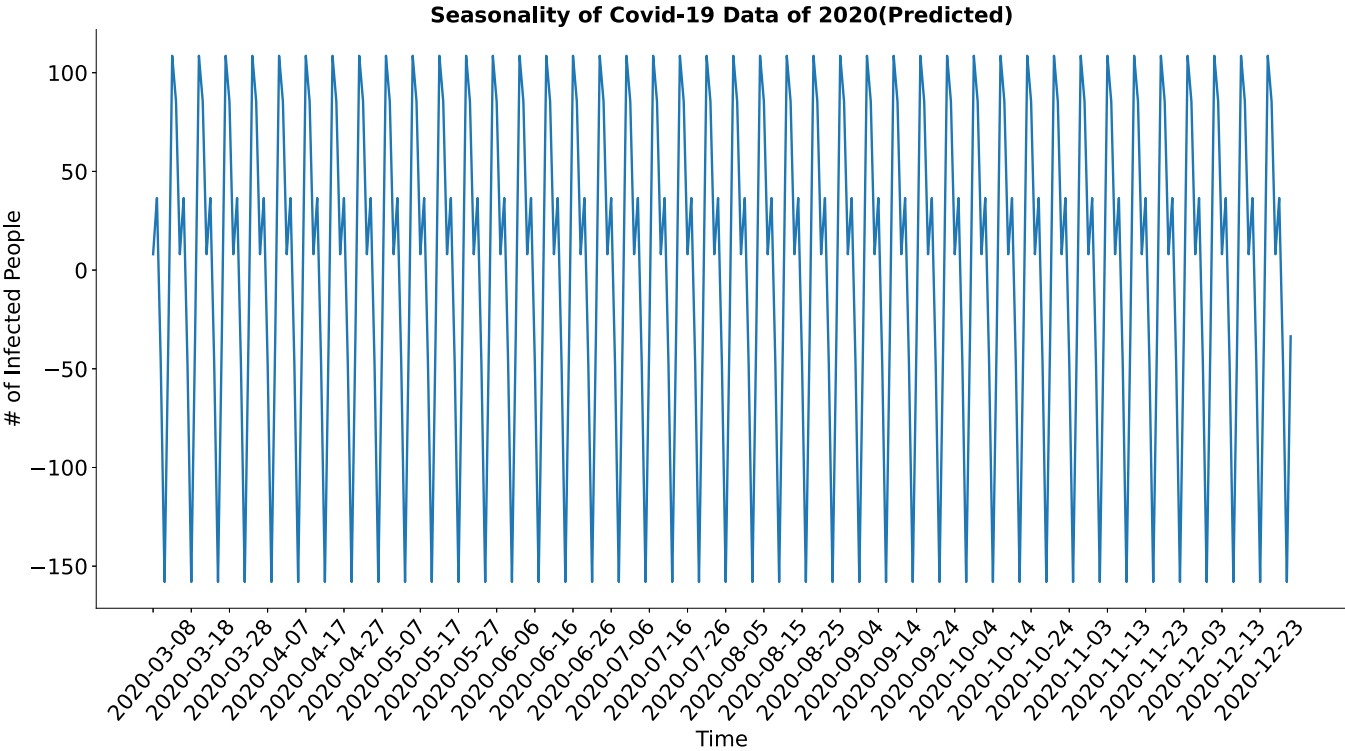

**Fig 21. Seasonality of Covid-19 data of 2020(Predicted).** Seasonality of the synthetic COVID-19 data.

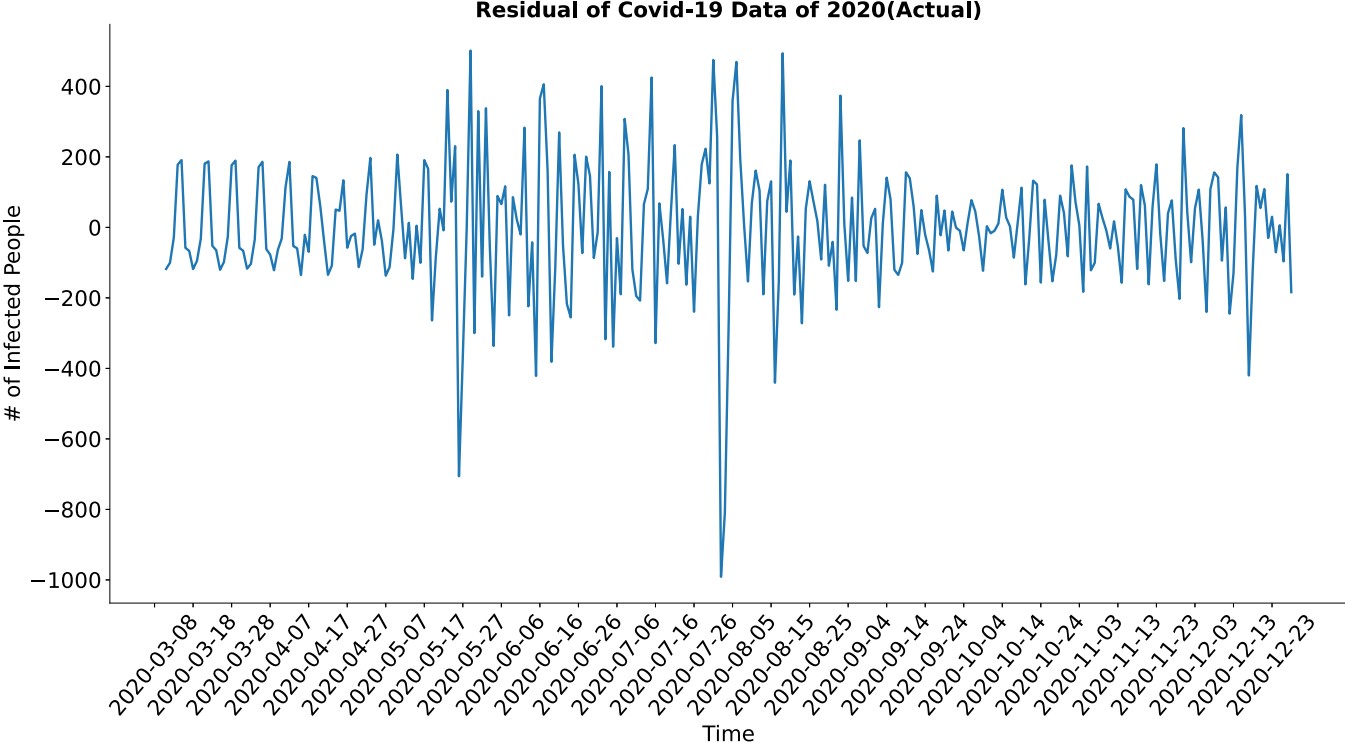

**Fig 22. Residual of Covid-19 data of 2020(Actual).** Residual of the actual COVID-19 data.

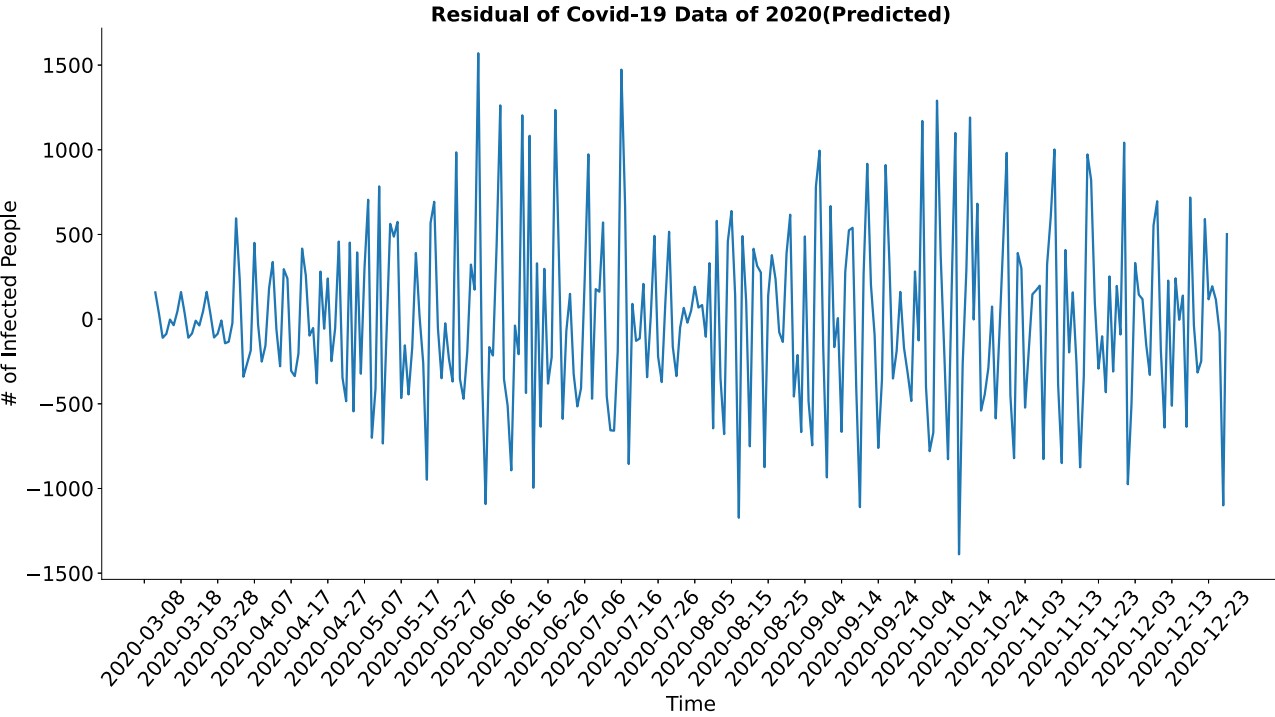

**Fig 23. Residual of Covid-19 data of 2020(Predicted).** Residual of the synthetic COVID-19 data.

To calculate the accuracy of the given forecast, we calculate the aforementioned error measures presented in Table 8.

The error measures are acceptable given the magnitude of the data, but there is room for improvement shall be demonstrated in the following subsection.

## Forecast on the synthesized data

The synthesized data is daily Dengue infection data of Bangladesh from 2010 to July 2022. Following Box-Jenkin's method, we firstly check for the stationarity of the data based on the Augmented Dicky Fuller (ADF) test. ADF test returns the value of -6.6531 with p-value = 0.01, which implies that the data is stationary.

We run multiple Fourier ARIMA models and calculate their AIC, AICc and BIC. The best model is chosen based on the minimum value of the criterion. We present 5 of the top results in Table 9. Here in each case of Fourier transformation, we used one pair of trigonometric terms where each pair is comprised of a sine and a cosine term as defined in (4) and the periodicity of the Fourier term is used to be 365.25. Prior to this we have used box cox transformation of $\lambda = 0.49$.

Here, the best model to use is ARIMA(7,0,7). We fit the given model, which gives us the coefficients in Table 10.

To check the goodness of fit of the model, we use the Ljung box test, which returns the p-value = 0.07749 > 0.05, i.e. we accept the null hypothesis: *"The model does not show lack ness of fit/ the residuals are not autocorrelated/ the residuals are random white noise"*.

Given everything in place, we forecast the infection for the rest of 2023, i.e. from August to December. The forecast is illustrated in the given figure (Fig 27).

## Box-Jenkin's Method of Model Selection

**Fig 24. Box-Jenkin's method of model selection.** Flow chart of Box-Jenkin's Method.

**Table 6. Selection of best model based on criteria for aggregated data.**

| Model | AIC | AICc | BIC |
|---|---|---|---|
| SARIMA$(1, 0, 0)(0, 1, 1)_{12}$ | 2603.57 | 2603.76 | 2612.22 |
| SARIMA$(1, 0, 0)(0, 1, 2)_{12}$ | 2604 | 2604.32 | 2615.53 |
| SARIMA$(1, 0, 0)(1, 1, 1)_{12}$ | 2604 | 2604.62 | 2615.84 |
| SARIMA$(1, 0, 1)(0, 1, 1)_{12}$ | 2604.36 | 2604.68 | 2615.89 |
| SARIMA$(2, 0, 0)(0, 1, 1)_{12}$ | 2604.42 | 2604.73 | 2615.95 |

The table depicts the AIC, AICc and BIC of the simulated SARIMA model for aggregated data and best model is selected based on the minimum value of the criterion.

**Table 7. Coefficients of SARIMA (1, 0, 0)(0, 1, 1)$_{12}$.**

|  | ar1 | sma1 |
|---|---|---|
|  | 0.5511 | -0.8546 |
| S.E. | 0.0721 | 0.0912 |

Coefficients of SARIMA (1, 0, 0)(0, 1, 1)$_{12}$ model to fit and forecast actual monthly data of Dengue infection in Bangladesh from 2010 to July, 2022. Here, ar implies autoregressive, SMA implies seasonal moving average, and the trailing number enumerates their coefficient ordering. SE implies the standard error of the mean.

To validate the goodness of the fit, we can analyze the model residual, illustrated in Fig 28. Here, the top graph is that of the residual with the timeline of the original data. The bottom left graph represents the Autocorrelation Function (ACF) with respect to the lag of the data. Almost all the values are within the significance e level, and the bottom right figure shows the distribution of the model's residuals. It implies that the residuals are distributed normally with zero mean.

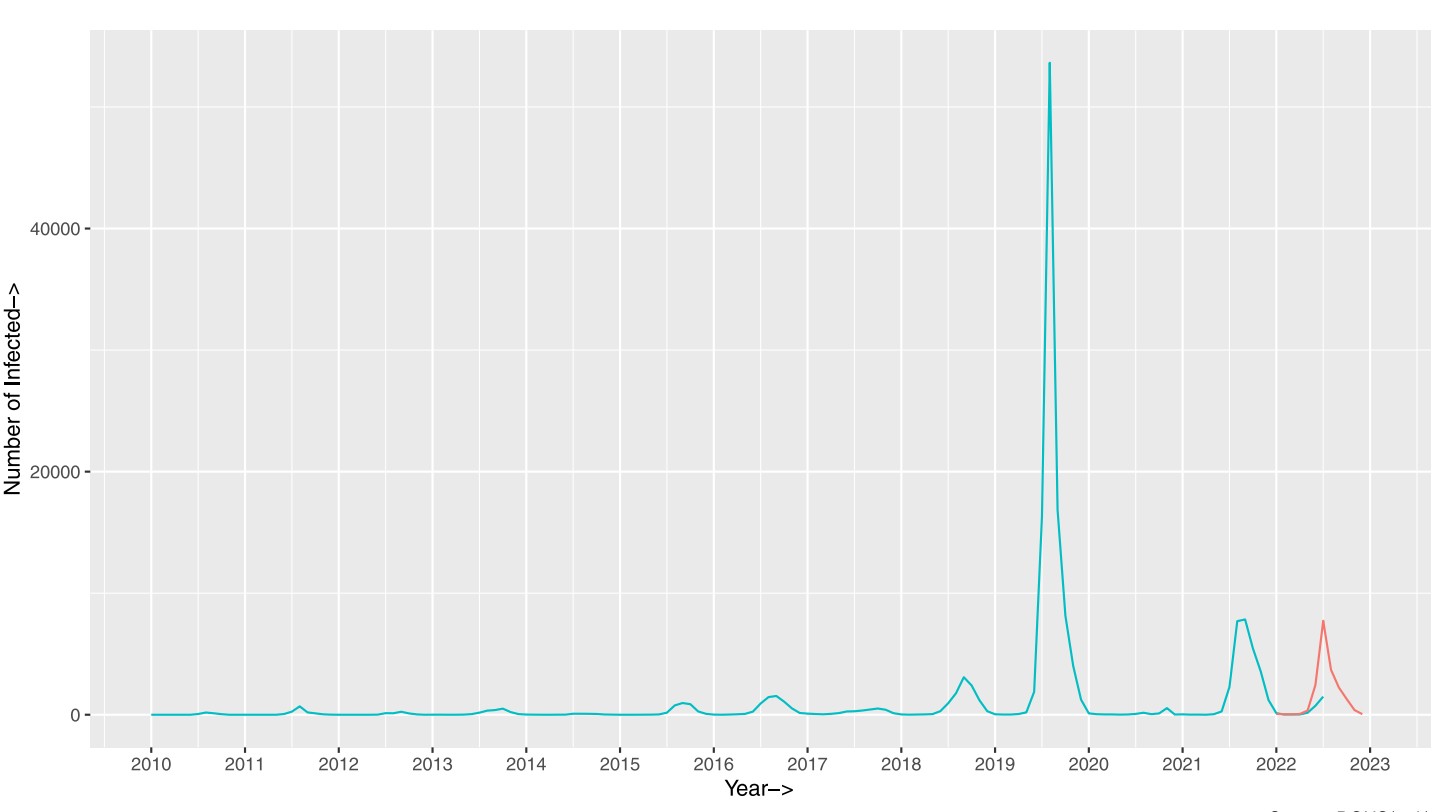

**Fig 25. Dengue infection forecast(Monthly).** The figure illustrates the forecast generated by SARIMA (1, 0, 0)(0, 1, 1)$_{12}$ from actual aggregated data.

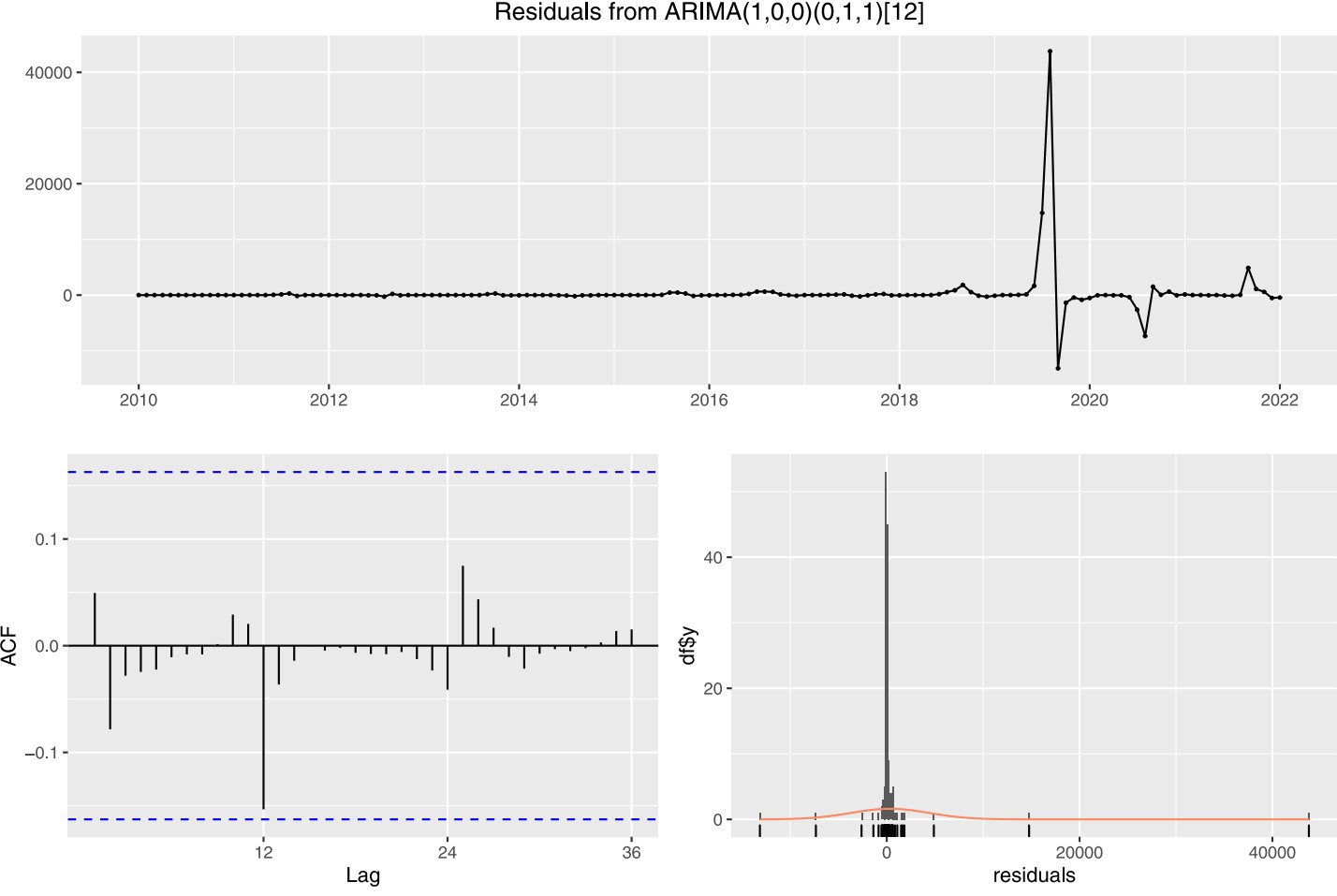

**Fig 26. Residual of the SARIMA (1,0,0,)(0,1,1) [12].** The figure illustrates in the bottom left graph that the ACF values for different choices of lag are all contained within the significance level (The dotted blue)and in the bottom right graph that the residuals are normally distributed with it's mean about 0.

To calculate the accuracy of the given forecast, we calculate the aforementioned error measures presented in Table 11.

The error measures are acceptable, given the magnitude of the data. In comparison to the error measures of the actual data illustrated in Table 8, we can see improvement in the Table 11. **Comparing the MASE term of the two tables shows about 72.76% decrement in error terms using the synthetic data over actual data.**

**Table 8. Error measures for the forecast of the SARIMA $(1, 0, 0)(0, 1, 1)_{12}$ of the actual aggregated data.**

| Data | RMSE | MAE | MASE |
|---|---|---|---|
| Monthly | 4092.712 | 753.6765 | 0.409654 |

The table depicts the error measures considered for the model. The errors are with in acceptable ranges given the magnitude of the data. The MASE is a point to be noted as it will used to compare and contrast the improvements achieved using synthetic data.

**Table 9. Selection of best model based on criteria.**

| Model | AIC | AICc | BIC |
|---|---|---|---|
| ARIMA(7,0,7) | 21711.25 | 21711.4 | 21827.03 |
| ARIMA(5,0,0) | 21819.04 | 21819.08 | 21876.94 |
| ARIMA(3,0,0) | 22147.88 | 22147.9 | 22192.91 |
| ARIMA(2,0,0) | 22527.02 | 22527.04 | 22565.61 |
| ARIMA(1,0,0) | 23476.24 | 23476.25 | 23508.4 |
| ARIMA(0,0,0) | 33245.98 | 33271.71 | 33271.71 |

The table depicts the AIC, AICc and BIC of the simulated Fourier ARIMA model for SDB generated data and best model is selected based on the minimum value of the criterion.

## Conclusion

In this paper, a novel temporal downscaling algorithm named Stochastic Bayesian Downscaling (SBD) algorithm has been proposed that can generate downscaled/deaggregated time series data of varying time length from the aggregated data. We have presented two case studies of Bangladesh using Dengue, 2022 data ranging from January to July and COVID-19, 2020 data to exhibit the workflow of the algorithm. In both case studies, the algorithm-generated synthetic data managed to replicate the mean of the actual data without ever being provided with it. In the case of the other statistical measures, the synthetic data could approximate it closely except for the maximum value. A way out of this issue is still an open question for research. Finally, we have tested how the classical statistical forecasting methods respond to the synthetic data with respect to actual aggregated data using monthly Dengue data of Bangladesh for the last 12 years. Our findings show that using synthetic data over actual aggregated data, we were able to reduce the scale-free error measure by **72.76%**.

The SBD algorithm presented in this paper is designed to handle integer data by imposing certain restrictions but can be generalized to handle real numbers upon relaxing such restrictions. Hence, exploring diverse use cases in public health, epidemiology, economics, and finance can be a future direction of research. In this paper, we have only studied how statistical forecasting models respond to synthetic data compared to actual data. Repeating similar

**Table 10. Coefficients of ARIMA(7,0,7).**

| | ar1 | ar2 | ar3 | ar4 | ar5 | ar6 | ar7 | ma1 |
|---|---|---|---|---|---|---|---|---|
| | -0.5273 | 0.3109 | 1.2946 | 1.0562 | 0.2775 | -0.6222 | -0.7940 | 0.8055 |
| S.E. | 0.0513 | 0.0310 | 0.0419 | 0.0755 | 0.0323 | 0.0353 | 0.0488 | 0.0471 |
| | ma2 | ma3 | ma4 | ma5 | ma6 | ma7 | intercept | s1−365 |
| | 0.0718 | -1.0032 | -1.1256 | -0.5327 | 0.3051 | 0.6454 | 3.3498 | 6.6197 |
| S.E. | 0.0340 | 0.0365 | 0.0602 | 0.0321 | 0.0356 | 0.0303 | 1.4789 | 1.6859 |
| | c1−365 | | | | | | | |
| | -0.7430 | | | | | | | |
| S.E. | 1.6857 | | | | | | | |

Coefficients of ARIMA(7,0,7) model to fit and forecast SDB generated data of Dengue infection in Bangladesh from 2010 to July 2022. Here, ar implies auto-regressive, ma implies the moving average, s and c represent the coefficient of the sine and cosine of Fourier term, intercept implies the constant term, and the trailing number enumerates their coefficient ordering. SE implies the standard error of the mean.

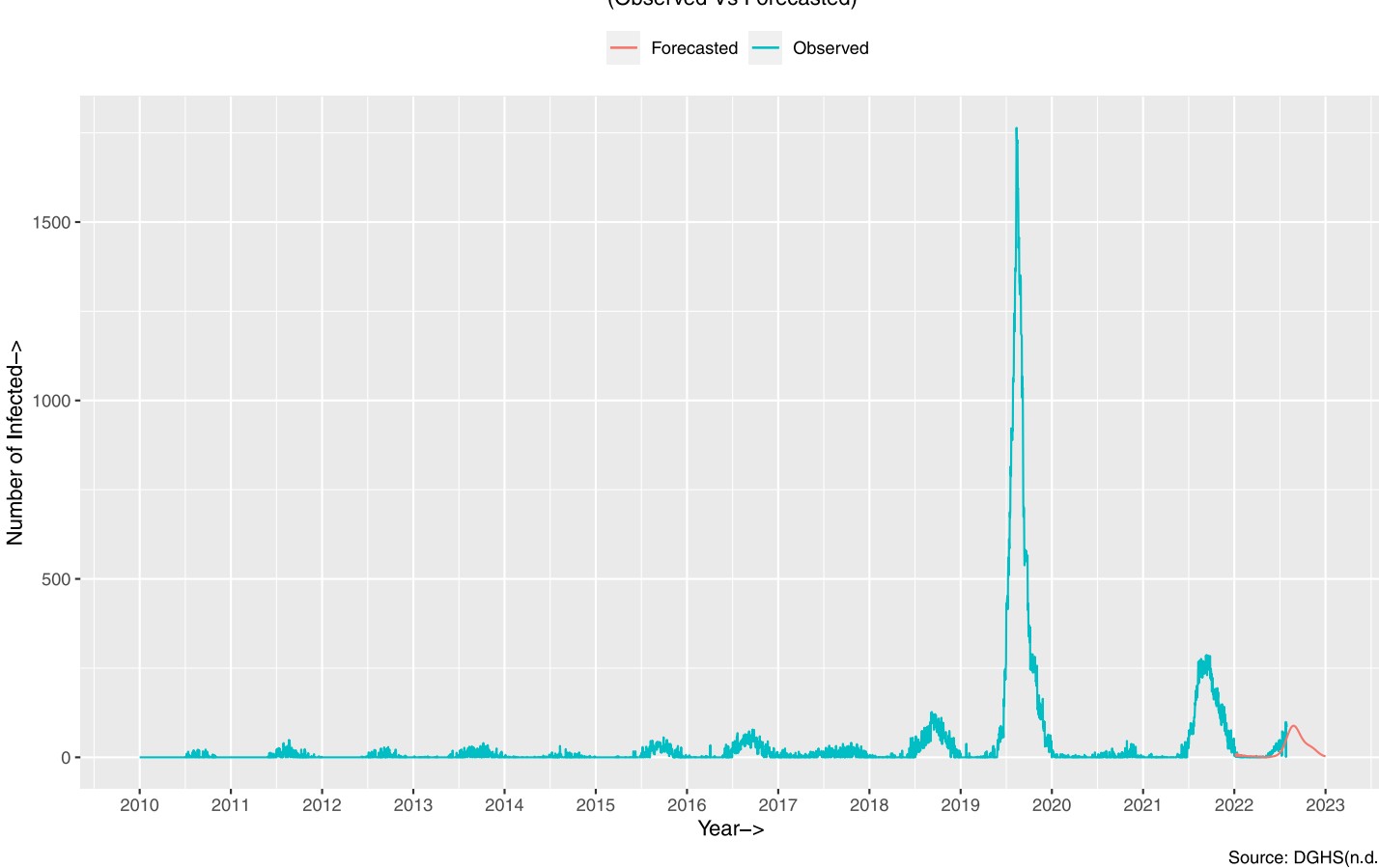

**Fig 27. Dengue infection forecast(Daily).** The figure illustrates the forecast generated by ARIMA(7,0,7) from actual aggregated data.

studies for the predictive class of machine learning models like Long Short Term Memory (LSTM), XGboost, etc is a further scope of research.

The downscaling algorithm has been predominantly used in geology to facilitate outputs of the prevalent models in the field. Very few applications have been made in epidemiology, and most of the application is spatial downscaling. This paper contributes to the current body of knowledge by proposing a parametric, probabilistic one-dimensional downscaling algorithm using aggregated data in the field of epidemiology that facilitates existing classical statistical forecasting tools box to generate better forecasts than the aggregated data. As we know, forecasting models like ARIMA and SARIMA are sensitive to data discontinuity and outliers, hence, the SBD algorithm can be implemented as pre model cleaning step to curate better results on a finer scale. As the SBD algorithm can increase *data volume to a significant scale (e.g. downscaling monthly data to daily data can increase the number of data points to 30 times on average) while preserving key statistics and properties of the data*, hence such downscaled data can open the avenue for exploration using state of the art neural network model which often requires large volume of data to generate fruitful outcome.

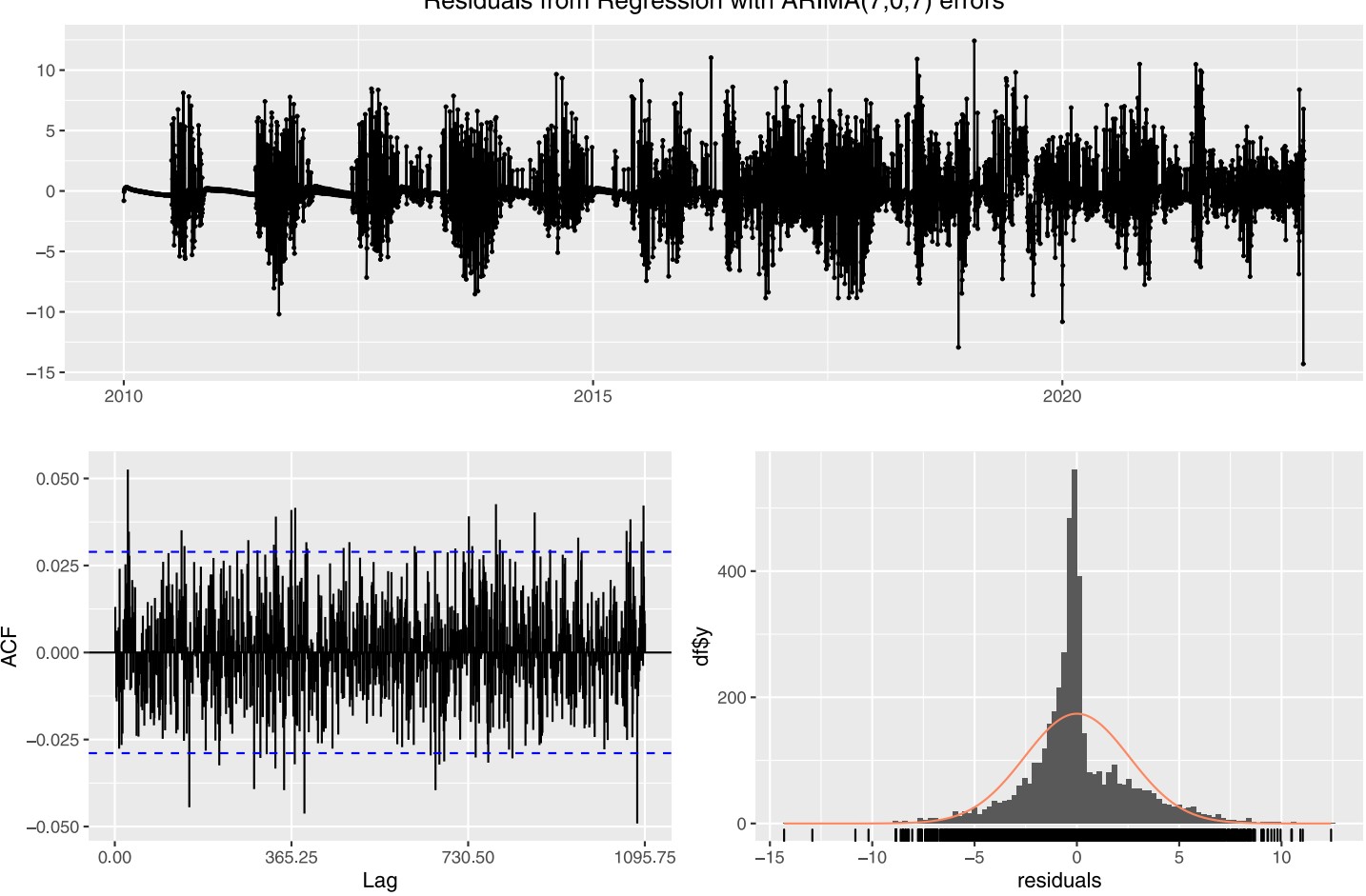

**Fig 28. Residuals from regression with ARIMA (7, 0, 7) errors.** The figure illustrates in the bottom left graph that the ACF values for different choices of lag are mostly contained within the significance level (The dotted blue)and in the bottom right graph that the residuals are normally distributed with it's mean about 0.

**Table 11. Error measures for the forecast of the ARIMA (7, 0, 7) of the synthetic daily data.**

| Data | RMSE | MAE | MASE |
|------|------|-----|------|
| Daily | 18.71255 | 6.593062 | 0.1115845 |

The table depicts the error measures considered for the model. The errors are with in acceptable ranges given the magnitude of the data. In comparision to the Table 8, **the scale independent error term, MASE shows 72.76% decrement**.

# Supporting information

**S1 Data.**
(CSV)

**S2 Data.**
(CSV)

**S3 Data.**
(CSV)

## Author Contributions

**Conceptualization:** Mahadee Al Mobin, Md. Kamrujjaman.

**Data curation:** Mahadee Al Mobin.

**Formal analysis:** Mahadee Al Mobin, Md. Kamrujjaman.

**Investigation:** Md. Kamrujjaman.

**Methodology:** Mahadee Al Mobin, Md. Kamrujjaman.

**Software:** Mahadee Al Mobin, Md. Kamrujjaman.

**Supervision:** Md. Kamrujjaman.

**Validation:** Md. Kamrujjaman.

**Writing – original draft:** Mahadee Al Mobin.

**Writing – review & editing:** Md. Kamrujjaman.

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
