## [Decision Letter · Decision Letter 0]

23 Oct 2023

PONE-D-23-27740Downscaling Epidemiological Time Series Data for Improving Forecasting Accuracy: An Algorithmic ApproachPLOS ONE

Dear Dr. Kamrujjaman,

Thank you for submitting your manuscript to PLOS ONE. After careful consideration, we feel that it has merit but does not fully meet PLOS ONE’s publication criteria as it currently stands. Therefore, we invite you to submit a revised version of the manuscript that addresses the points raised during the review process.

We look forward to receiving your revised manuscript.

Kind regards,

Salim Heddam

Academic Editor

PLOS ONE

Journal Requirements:

Additional Editor Comments:

Reviewer 1:Article is very well prepared and structed .But few minor corrections required:

(1)Please add some projection values in abstract Section.

(2)Please mention the link of data source here.

My additional comment given in attached file.

Reviewer 2:The paper does not provide much novelty. The authors should explain the proposed technique in detail.

Major portion of the paper deals with usual time series models like ARIMA and SARIMA. The description of accuracy statistics is in much details while it is available in literature and researchers are well known about it.

It may de precisely presented.

The conclusion and abstract should be revised with specific objectives.

Reviewers' comments:

Reviewer's Responses to Questions

**Comments to the Author**

1. Is the manuscript technically sound, and do the data support the conclusions?

Reviewer #1: Yes

Reviewer #2: No

2. Has the statistical analysis been performed appropriately and rigorously? 

Reviewer #1: Yes

Reviewer #2: No

3. Have the authors made all data underlying the findings in their manuscript fully available?

Reviewer #1: Yes

Reviewer #2: No

4. Is the manuscript presented in an intelligible fashion and written in standard English?

Reviewer #1: Yes

Reviewer #2: Yes

5. Review Comments to the Author

Reviewer #1: Article is very well prepared and structed .But few minor corrections required:

(1)Please add some projection values in abstract Section.

(2)Please mention the link of data source here.

My additional comment given in attached file.

Reviewer #2: The paper does not provide much novelty. The authors should explain the proposed technique in detail.

Major portion of the paper deals with usual time series models like ARIMA and SARIMA. The description of accuracy statistics is in much details while it is available in literature and researchers are well known about it.

It may de precisely presented.

The conclusion and abstract should be revised with specific objectives.

6. PLOS authors have the option to publish the peer review history of their article (what does this mean?). If published, this will include your full peer review and any attached files.

Reviewer #1: No

Reviewer #2: No

---

## [Author Response · Author response to Decision Letter 0]

13 Nov 2023

The Respond to Reviewers file is uploaded in the file section. File name-Response to Reviewers_R1_PONE-D-23-27740

---

## [Decision Letter · Decision Letter 1]

30 Nov 2023

Downscaling Epidemiological Time Series Data for Improving Forecasting Accuracy: An Algorithmic Approach

PONE-D-23-27740R1

Dear Dr. Kamrujjaman

We’re pleased to inform you that your manuscript has been judged scientifically suitable for publication and will be formally accepted for publication once it meets all outstanding technical requirements.

Kind regards,

Salim Heddam

Academic Editor

PLOS ONE

Additional Editor Comments (optional):

Reviewer 1#:Article is accepted. The article is well-prepared and makes a significant contribution to the field.

Reviewer 2#:The revised manuscript looks good. It is suggested to write few lines on future scope in the conclusion section. Best of luck.

Reviewers' comments:

Reviewer's Responses to Questions

**Comments to the Author**

1. If the authors have adequately addressed your comments raised in a previous round of review and you feel that this manuscript is now acceptable for publication, you may indicate that here to bypass the “Comments to the Author” section, enter your conflict of interest statement in the “Confidential to Editor” section, and submit your "Accept" recommendation.

Reviewer #1: All comments have been addressed

Reviewer #2: All comments have been addressed

2. Is the manuscript technically sound, and do the data support the conclusions?

Reviewer #1: Yes

Reviewer #2: (No Response)

3. Has the statistical analysis been performed appropriately and rigorously? 

Reviewer #1: Yes

Reviewer #2: (No Response)

4. Have the authors made all data underlying the findings in their manuscript fully available?

Reviewer #1: Yes

Reviewer #2: (No Response)

5. Is the manuscript presented in an intelligible fashion and written in standard English?

Reviewer #1: Yes

Reviewer #2: (No Response)

6. Review Comments to the Author

Reviewer #1: Article is accepted. The article is well-prepared and makes a significant contribution to the field.

Reviewer #2: The revised manuscript looks good. It is suggested to write few lines on future scope in the conclusion section. Best of luck.

7. PLOS authors have the option to publish the peer review history of their article (what does this mean?). If published, this will include your full peer review and any attached files.

Reviewer #1: No

Reviewer #2: No

---

## [Editor Report · Acceptance letter]

4 Dec 2023

PONE-D-23-27740R1 

Downscaling Epidemiological Time Series Data for Improving Forecasting Accuracy: An Algorithmic Approach 

Dear Dr. Kamrujjaman:

I'm pleased to inform you that your manuscript has been deemed suitable for publication in PLOS ONE. Congratulations! Your manuscript is now with our production department. 

Kind regards, 

on behalf of

Dr. Salim Heddam 

Academic Editor

PLOS ONE